# Cortical activity during naturalistic music listening reflects short-range predictions based on long-term experience

Pius Kern, Micha Heilbron, Floris P de Lange, Eelke Spaak*

Radboud University Nijmegen, Donders Institute for Brain, Cognition and Behaviour, Nijmegen, Netherlands

**Abstract** Expectations shape our experience of music. However, the internal model upon which listeners form melodic expectations is still debated. Do expectations stem from Gestalt-like principles or statistical learning? If the latter, does long-term experience play an important role, or are short-term regularities sufficient? And finally, what length of context informs contextual expectations? To answer these questions, we presented human listeners with diverse naturalistic compositions from Western classical music, while recording neural activity using MEG. We quantified note-level melodic surprise and uncertainty using various computational models of music, including a state-of-the-art transformer neural network. A time-resolved regression analysis revealed that neural activity over fronto-temporal sensors tracked melodic surprise particularly around 200ms and 300–500ms after note onset. This neural surprise response was dissociated from sensory-acoustic and adaptation effects. Neural surprise was best predicted by computational models that incorporated long-term statistical learning—rather than by simple, Gestalt-like principles. Yet, intriguingly, the surprise reflected primarily short-range musical contexts of less than ten notes. We present a full replication of our novel MEG results in an openly available EEG dataset. Together, these results elucidate the internal model that shapes melodic predictions during naturalistic music listening.

*For correspondence:
eelke.spaak@donders.ru.nl

## Editor's evaluation

This study models the predictions a listener makes in music in two ways: how different model algorithms compare in their performance at predicting the upcoming notes in a melody, and how well they predict listeners' brain responses to these notes. The study will be important as it implements and compares three contemporary models of music prediction. In a set of convincing analyses, the authors find that musical melodies are best predicted by models taking into account long-term experience of musical melodies, whereas brain responses are best predicted by applying these models to only a few most recent notes.

## Introduction

The second movement of Haydn's symphony No. 94 begins with a string section creating the expectation of a gentle and soft piece, which is suddenly interrupted by a tutti fortissimo chord. This startling motif earned the composition the nickname 'Surprise symphony'. All music, in fact, plays with listeners' expectations to evoke musical enjoyment and emotions, albeit often in more subtle ways (*Huron, 2006*; *Juslin and Västfjäll, 2008*; *Meyer, 1957*; *Salimpoor et al., 2015*). A central element of music which induces musical expectations is melody, the linear sequence of notes alternating in pitch. Within a musical piece and style, such as Western classical music, certain melodic patterns appear more frequently than others, establishing a musical syntax (*Krumhansl, 2015*; *Patel, 2003*;

*Rohrmeier et al., 2011*). Human listeners have been proposed to continuously form predictions on how the melody will continue based on these regularities (*Koelsch et al., 2019*; *Meyer, 1957*; *Tillmann et al., 2014*; *Vuust et al., 2022*).

In support of prediction-based processing of music, it has been shown that listeners are sensitive to melodic surprise. Behaviorally, higher surprise notes are rated as more unexpected (*Krumhansl and Kessler, 1982*; *Marmel et al., 2008*; *Marmel et al., 2010*; *Pearce et al., 2010*; *Schmuckler, 1989*) and impair performance, for example in dissonance detection tasks (*Pearce et al., 2010*; *Sears et al., 2019*). Listeners continue melodic primes with low-surprise notes in musical cloze tasks (*Carlsen, 1981*; *Morgan et al., 2019*; *Schmuckler, 1989*). Neural activity tracks melodic surprise (*Di Liberto et al., 2020*) and high-surprise notes elicit electrophysiological signatures indicative of surprise processing, in particular the mismatch negativity (*Brattico et al., 2006*; *Mencke et al., 2021*; *Näätänen et al., 2007*; *Quiroga-Martinez et al., 2020*) and P3 component (*Quiroga-Martinez et al., 2020*) (for a review see *Koelsch et al., 2019*), but also the P2 component (*Omigie et al., 2013*), a late negative activity around 400ms (*Miranda and Ullman, 2007*; *Pearce et al., 2010*), and oscillatory activity (*Omigie et al., 2019*; *Pearce et al., 2010*). Despite this extensive body of neural and behavioral evidence on the effects of melodic expectations in music perception, the form and content of the internal model generating these expectations remain unclear. Furthermore, the evidence stems primarily from studying the processing of relatively artificial stimuli, and how these findings extend to a more naturalistic setting is unknown.

We set out to answer three related open questions regarding the nature of melodic expectations, as reflected in neural activity. First, are expectations best explained by a small set of Gestalt-like principles (*Krumhansl, 2015*; *Narmour, 1990*; *Narmour, 1992*; *Temperley, 2008*; *Temperley, 2014*), or are they better captured by statistical learning (*Pearce, 2005*; *Pearce and Wiggins, 2012*; *Rohrmeier and Koelsch, 2012*)? According to Gestalt-like models, expectations stem from relatively simple rules also found in music theory, for example that intervals between subsequent notes tend to be small. From a statistical learning perspective, in contrast, listeners acquire internal predictive models, capturing potentially similar or different principles, through exposure to music. Overall, statistical learning models have proven slightly better fits for musical data (*Temperley,*

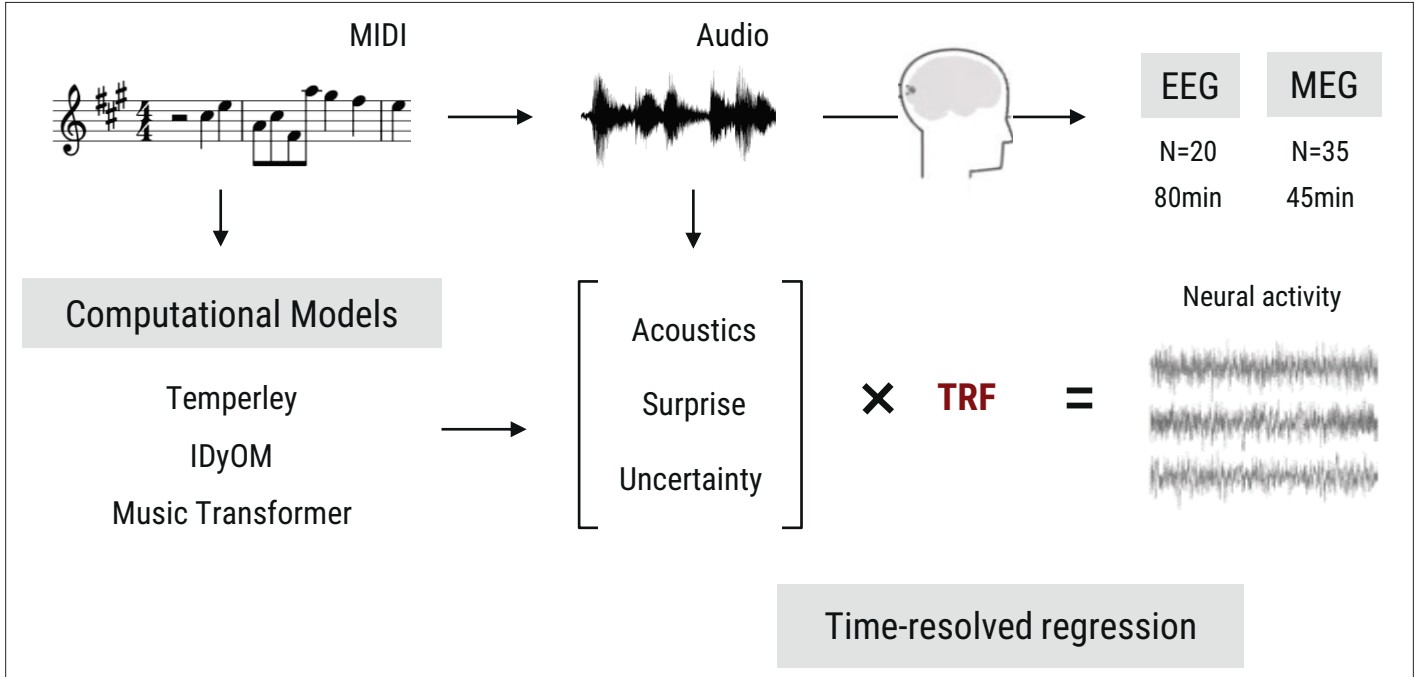

**Figure 1.** Overview of the research paradigm. Listeners undergoing EEG (data from *Di Liberto et al., 2020*) or MEG measurement (novel data acquired for the current study) were presented with naturalistic music synthesized from MIDI files. To model melodic expectations, we calculated note-level surprise and uncertainty estimates via three computational models reflecting different internal models of expectations. We estimated the regression evoked response or temporal response function (TRF) for different features using time-resolved linear regression on the M|EEG data, while controlling for low-level acoustic factors.

*2014*) and for human listeners' expectations assessed behaviorally (*Morgan et al., 2019*; *Pearce and Wiggins, 2006*; *Temperley, 2014*), but the two types of models have rarely been directly compared. Second, if statistical learning drives melodic expectations, does this rely on long-term exposure to music, or might it better reflect the local statistical structure of a given musical piece? Finally, independent of whether melodic expectations are informed by short or long-term *experience*, we ask how much temporal *context* is taken into account by melodic expectations; that is whether these are based on a short- or a longer-range context. On the one hand, the brain might use as much temporal context as possible in order to predict optimally. On the other hand, the range of echoic memory is limited and temporal integration windows are relatively short, especially in sensory areas (*Hasson et al., 2008*; *Honey et al., 2012*; *Himberger et al., 2018*). Therefore, melodic expectations could be based on shorter-range context than would be statistically optimal. To address this question, we derived model-based probabilistic estimates of expectations using the Music Transformer (*Huang et al., 2018*). This is a state-of-the-art neural network model that can take long-range (and variable) context into account much more effectively than the n-gram models previously used to model melodic expectations, since transformer models process blocks of (musical) context as a whole, instead of focusing on (note) sequences of variable, yet limited, length.

In the current project, we approached this set of questions as follows (*Figure 1*). First, we operationalized different sources of melodic expectations by simulating different predictive architectures: the Probabilistic Model of Melody Perception (*Temperley, 2008*; *Temperley, 2014*), which is a Gestalt-like model; the Information Dynamics of Music (IDyOM) model, an n-gram based statistical learning model (*Pearce, 2005*; *Pearce and Wiggins, 2012*); and the aforementioned Music Transformer. We compared the different computational models' predictive performance on music data to establish them as different hypotheses about the sources of melodic expectations. We then analyzed a newly acquired MEG dataset obtained while participants (n=35) were listening to diverse, naturalistic, musical stimuli using time-resolved regression analysis. This allowed us to disentangle the contributions of different sources of expectations, as well as different lengths of contextual information, to the neural signature of surprise processing that is so central to our experience of music. To preview our results: we found that melodic surprise strongly modulates the evoked response, and that this effect goes beyond basic acoustic features and simple repetition effects, confirming that also in naturalistic music listening, brain responses are shaped by melodic expectations. Critically, we found that neural melodic surprise is best captured by long-term statistical learning; yet, intriguingly, depends primarily on short-range musical context. In particular, we observed a striking dissociation at a context window of about ten notes: models taking longer-range context into account become better at predicting music, but worse at predicting neural activity. Superior temporal cortical sources most strongly contributed to the surprise signature, primarily around 200ms and 300–500ms after note onset. Finally, we present a full replication of our findings in an independent openly available EEG dataset (*Di Liberto et al., 2020*).

## Results
### Music analysis

We quantified the note-level surprise and uncertainty using different computational models of music, which were hypothesized to capture different sources of melodic expectation (see Materials and methods for details). The Probabilistic Model of Melody Perception (Temperley) (*Temperley, 2008*; *Temperley, 2014*) rests on a few principles derived from musicology and thus represents Gestalt-like perception (*Morgan et al., 2019*). The Information Dynamics of Music (IDyOM) model (*Pearce and Wiggins, 2012*) captures expectations from statistical learning, either based on short-term regularities in the current musical piece (IDyOM stm), long-term exposure to music (IDyOM ltm), or a combination of the former two (IDyOM both). The Music Transformer (MT) (*Huang et al., 2018*) is a state-of-the-art neural network model, which also reflects long-term statistical learning but is more sensitive to longer-range structure. In a first step, we aimed to establish the different models as distinct hypotheses about the sources of melodic expectations. We examined how well the models predicted music data and to what extent their predictions improved when the amount of available context increased.

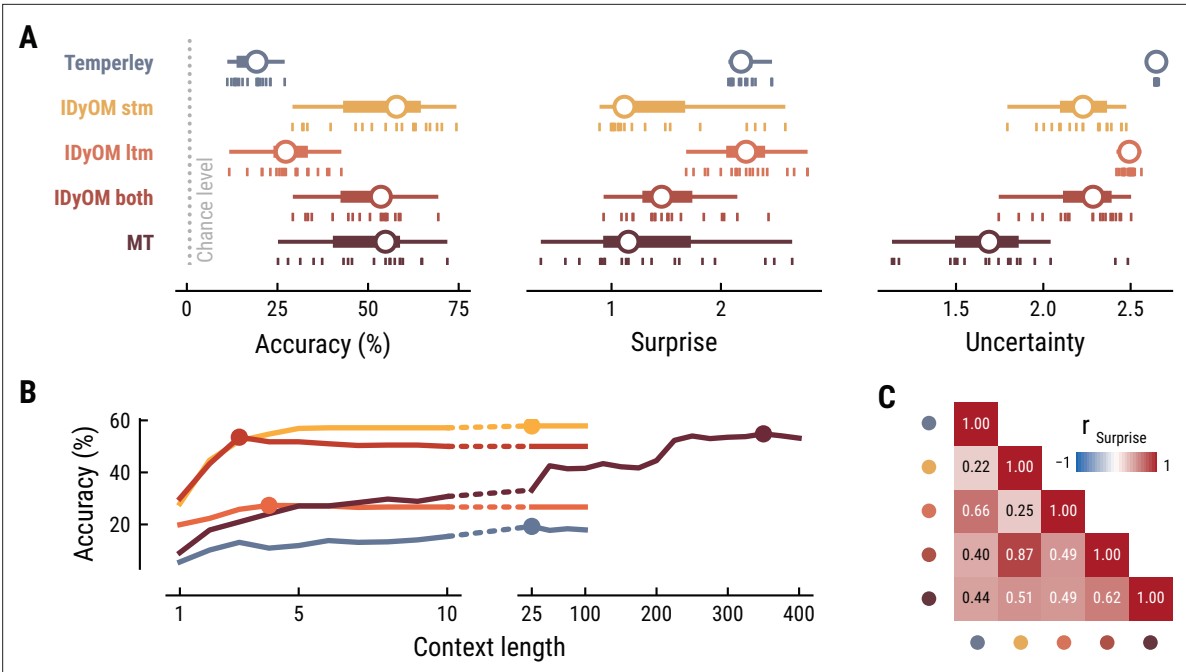

**Figure 2.** Model performance on the musical stimuli used in the MEG study. (**A**) Comparison of music model performance in predicting upcoming note pitch, as composition-level accuracy (left; higher is better), median surprise across notes (middle; lower is better), and median uncertainty across notes (right). Context length for each model is the best performing one across the range shown in (**B**). Vertical bars: single compositions, circle: median, thick line: quartiles, thin line: quartiles ±1.5 × interquartile range. (**B**) Accuracy of note pitch predictions (median across 19 compositions) as a function of context length and model class (same color code as (**A**)). Dots represent maximum for each model class. (**C**) Correlations between the surprise estimates from the best models. (For similar results for the musical stimuli used in the EEG study, see **Appendix 1—figure 2**).

## IDyOM stm and Music Transformer show superior melodic prediction

First, we tested how well the different computational models predicted the musical stimuli presented in the MEG study (**Figure 2**). Specifically, we quantified the accuracy with which the models predicted upcoming notes, given a certain number of previous notes as context information. While all models performed well above chance level accuracy (1/128=0.8%), the IDyOM stm (median accuracy across compositions: 57.9%), IDyOM both (53.5%), and Music Transformer (54.8%) models performed considerably better than the Temperley (19.3%) and IDyOM ltm (27.3%) models, in terms of median accuracy across compositions (**Figure 2A** left). This pattern was confirmed in terms of the models' note-level surprise, which is a continuous measure of predictive performance. Here lower values indicate a better ability to predict the next note given the context (median surprise across compositions: Temperley = 2.18, IDyOM stm = 1.12, IDyOM ltm = 2.23, IDyOM both = 1.46, MT = 1.15, **Figure 2A** middle). The median surprise is closely related to the cross-entropy loss, which can be defined as the mean surprise across all notes (Temperley = 2.7, IDyOM stm = 2, ltm = 2.47, both = 1.86, Music Transformer = 1.81). Furthermore, the uncertainty, defined as the entropy of the probability distribution at each time point, characterizes each model's confidence (inverse) in its predictions (maximum uncertainty = 4.85 given a uniform probability distribution). The Music Transformer model formed predictions more confidently than the other models, whereas the Temperley model displayed the highest uncertainty (median uncertainty across compositions: Temperley = 2.65, IDyOM stm = 2.23, ltm = 2.49, both = 2.28, MT = 1.69, **Figure 2A** right). Within the IDyOM class, the stm model consistently showed lower uncertainty compared to the ltm model, presumably reflecting a greater consistency of melodic patterns within versus across compositions. As a result, the both model was driven by the stm model, since it combines the ltm and stm components weighted by their uncertainty (mean stm weight = 0.72, mean ltm weight = 0.18).

## Music Transformer utilizes long-range musical structure

Next, we examined to what extent the different models utilize long-range structure in musical compositions or rely on short-range regularities by systematically varying the context length $k$ (above we

considered each model at its optimal context length, defined by the maximum accuracy). The Music Transformer model proved to be the only model for which the predictive accuracy increased considerably as the context length increased, from about 9.17% ($k$=1) up to 54.82% ($k$=350) (*Figure 2B*). The IDyOM models' performance, in contrast, plateaued early at context lengths between three and five notes (optimal $k$: stm: 25, ltm: 4, both: 3), reflecting the well-known sparsity issue of n-gram models (*Jurafsky and Martin, 2000*). Although the Temperley model benefited from additional musical context slightly, the increment was small and the accuracy was lower compared to the other models across all context lengths (5.58% at $k$=1 to 19.25% at $k$=25).

## Computational models capture distinct sources of musical expectation

To further evaluate the differences between models, we tested how strongly their surprise estimates were correlated across all notes in the stimulus set (*Figure 2C*). Since the IDyOM stm model dominated the both model, the two were correlated most strongly ($r$=.87). The lowest correlations occurred between the IDyOM stm on the one hand and the IDyOM ltm ($r$=0.24) and Temperley model ($r$=0.22) on the other hand. Given that all estimates quantified surprise, positive correlations of medium to large size were expected. More importantly, the models appeared to pick up substantial unique variance, in line with the differences in predictive performance explored above.

Taken together, these results established that the computational models of music capture different sources of melodic expectation. Only the Music Transformer model was able to exploit long-range structure in music to facilitate predictions of note pitch. Yet, short-range regularities in the current musical piece alone enabled accurate melodic predictions already: the IDyOM stm model performed remarkably well, even compared to the much more sophisticated Music Transformer. We confirmed these results on the musical stimuli from the EEG study (*Appendix 1—figure 2*).

## M|EEG analysis

We used a time-resolved linear regression approach (see Materials and methods for details) to analyse listeners' M|EEG data. By comparing different regression models, we asked (1) whether there is evidence for the neural processing of melodic surprise and uncertainty during naturalistic music listening and (2) which sources of melodic expectations, represented by the different computational models, best capture that. We quantified the performance of each regression model in explaining the MEG data by computing the correlation r between predicted and observed neural data. Importantly, we estimated r using fivefold cross-validation, thereby ruling out any trivial increase in predictive performance due to increases in number of regressors (i.e. free parameters).

The simplest model, the Onset model, contained a single regressor coding note onsets in binary fashion. Unsurprisingly, this model significantly explained variance in the recorded MEG data (mean r across participants = 0.12, SD = 0.03; one-sample t-test versus zero, $t_{34}$=25.42, p=1.06e-23, d=4.36, *Figure 3A* top left), confirming that our regression approach worked properly. The Baseline model included the note onset regressor, and additionally a set of regressors to account for sensory-acoustic features, such as loudness, sound type, pitch class (low/high), as well as note repetitions to account for sensory adaptation (*Auksztulewicz and Friston, 2016*; *Todorovic and de Lange, 2012*). The Baseline model explained additional variance beyond the Onset model ($\Delta r_{Baseline-Onset}$=0.013, SD = 0.006; paired-sample t-test, $t_{34}$=12.07, p=7.58e-14, d=2.07, *Figure 3A* bottom left), showing that differences in acoustic features and repetition further modulated neural activity elicited by notes.

## Long-term statistical learning best explains listeners' melodic surprise

We next investigated to which degree the surprise estimates from the different computational models of music could explain unique variance in the neural data, over and above that already explained by the Baseline model. All models performed significantly better than the Baseline model, providing evidence for tracking of neural surprise during naturalistic music listening (Temperley: $\Delta r_{Surprise-Baseline}$=0.002, SD = 0.001, paired-sample t-test, $t_{34}$=8.76 p=2.42e-09, d=1.5; IDyOM stm: $\Delta r_{Surprise-Baseline}$=0.001, SD = 0.001, $t_{34}$=5.66 p=9.39e-06, d=0.97; IDyOM ltm: $\Delta r_{Surprise-Baseline}$=0.003, SD = 0.002, $t_{34}$=12.74 p=2.51e-13, d=2.19; IDyOM both: $\Delta r_{Surprise-Baseline}$=0.002, SD = 0.001, $t_{34}$=8.77, p=2.42e-09, d=1.5; and Music Transformer: $\Delta r_{Surprise-Baseline}$=0.004, SD = 0.002, $t_{34}$=10.82, p=1.79e-11, d=1.86, corrected for multiple comparisons using the Bonferroni-Holm method) (*Figure 3A* right). Importantly, the Music Transformer and IDyOM ltm model significantly outperformed the other models (paired-sample t-test, MT-Temperley:

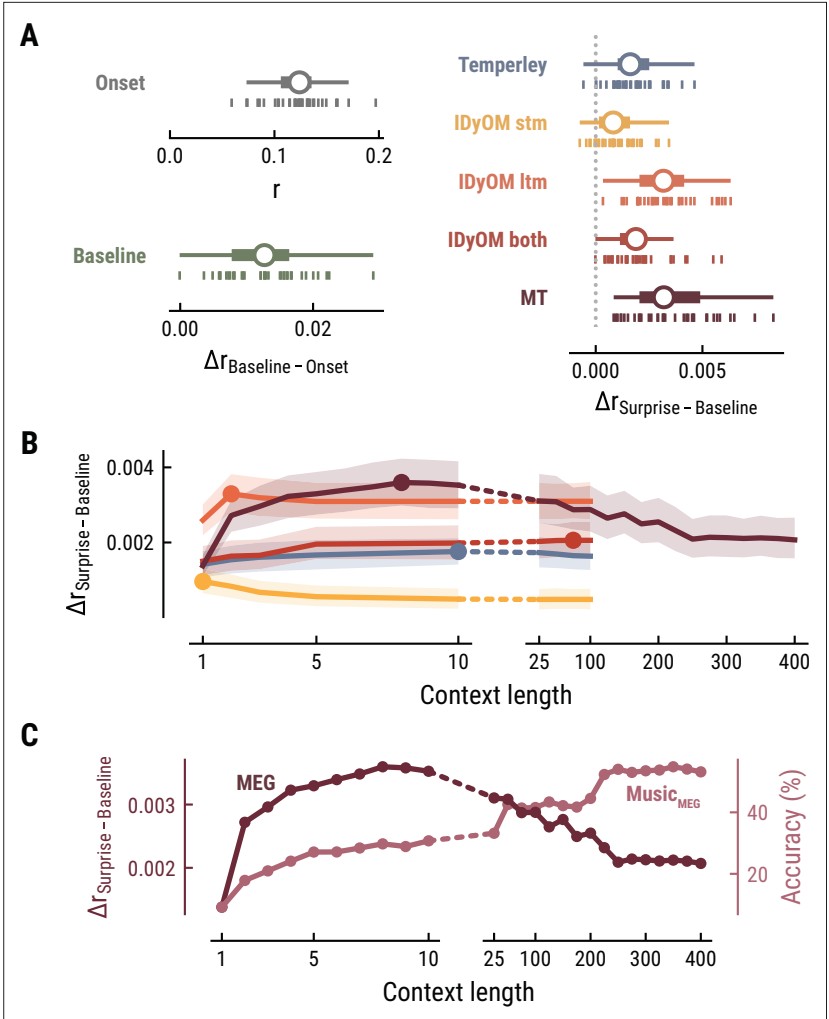

**Figure 3.** Model performance on MEG data from 35 listeners. (**A**) Cross-validated r for the Onset only model (top left). Difference in cross-validated r between the Baseline model including acoustic regressors and the Onset model (bottom left). Difference in cross-validated r between models including surprise estimates from different model classes (color-coded) and the Baseline model (right). Vertical bars: participants; box plot as in *Figure 2*. (**B**) Comparison between the best surprise models from each model class as a function of context length. Lines: mean across participants, shaded area: 95% CI. (**C**) Predictive performance of the Music Transformer (MT) on the MEG data (left y-axis, dark, mean across participants) and the music data from the MEG study (right y-axis, light, median across compositions).

$t_{34}=7.56$, $p=5.33e-08$, $d=1.30$; MT-IDyOM stm: $t_{34}=9.51$, $p=4.12e-10$, $d=1.63$, MT-IDyOM both: $t_{34}=8.87$, $p=2.07e-09$, $d=1.52$), with no statistically significant difference between the two (paired-sample t-test, $t_{34}=1.634$, $p=0.225$), whereas the IDyOM stm model performed worst. This contrasts with the music analysis, where the IDyOM stm model performed considerably better than the IDyOM ltm model. These observations suggest that listeners' melodic surprise is better explained by musical enculturation (i.e., exposure to large amounts of music across the lifetime), modeled as statistical learning on a large corpus of music (IDyOM ltm and MT), rather than by statistical regularities within the current musical piece alone (IDyOM stm) or Gestalt-like rules (Temperley).

## Short-range musical context shapes listeners' melodic surprise

We again systematically varied the context length *k* to probe which context length captures listeners' melodic surprise best (above we again considered each model at its optimal context length, defined by the maximum $\Delta r_{Surprise-Baseline}$ averaged across participants). The Temperley and IDyOM models' incremental predictive contribution were marginally influenced by context length, with early peaks

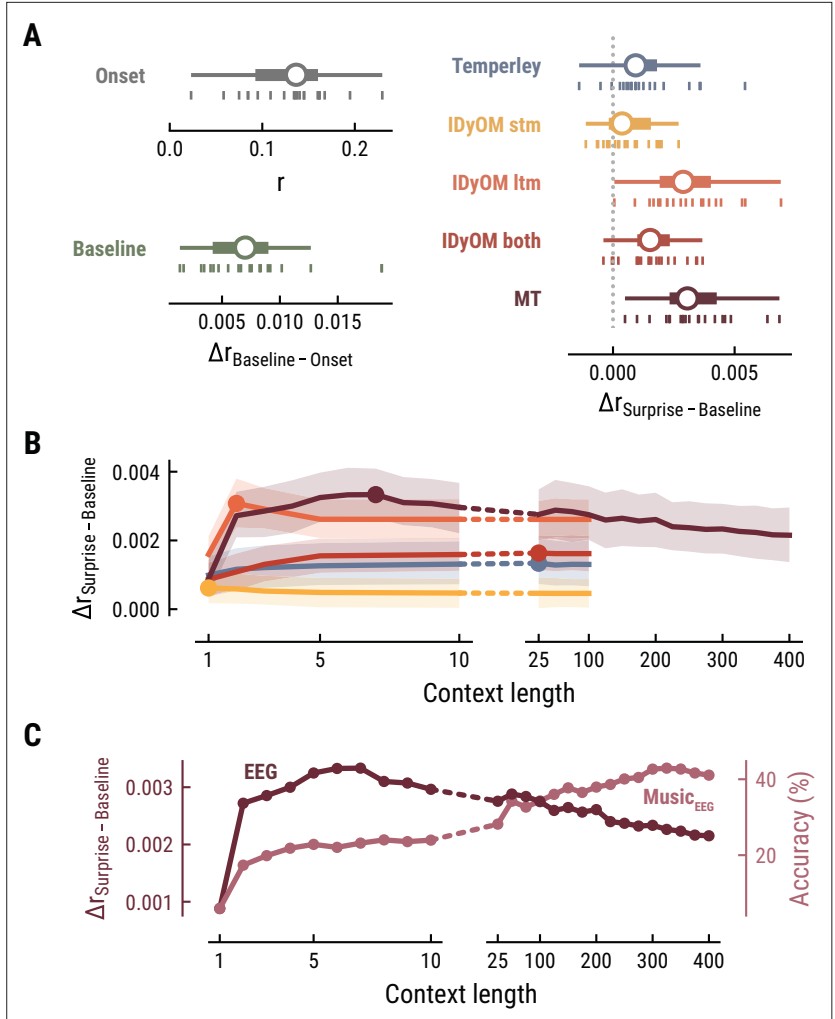

**Figure 4.** Model performance on EEG data from 20 listeners. All panels as in *Figure 3*, but applied to the EEG data and its musical stimuli.

for the IDyOM stm (*k*=1) and ltm (*k*=2) and later peaks for the both (*k*=75) and Temperley models (*k*=10) (*Figure 3B*). The roughly constant level of performance was expected based on the music analysis, since these models mainly relied on short-range context and their estimates of surprise were almost constant. In contrast, we reported above that the Music Transformer model extracts long-range structure in music, with music-predictive performance increasing up to context lengths of 350 notes. Strikingly, however, surprise estimates from the MT predicted MEG data best at a context length of nine notes and decreased for larger context lengths, even below the level of shorter ones (<10) (*Figure 3C*).

Together, these findings suggest that long-term experience of listeners (IDyOM ltm and MT) better captures neural correlates of melodic surprise than short-term statistical regularities (IDyOM stm). Yet, melodic expectations based on statistical learning might not necessarily rest on long-range temporal structure but rather shorter time scales between 5 and 10 notes. These results were replicated on the EEG data (*Figure 4*).

## Spatiotemporal neural characteristics of melodic surprise

To elucidate the spatiotemporal neural characteristics of naturalistic music listening, we further examined the temporal response functions (TRFs; or 'regression evoked responses') from the best model (MEG: MT at *k*=8, *Figure 5*; EEG: MT at *k*=7, *Figure 6*). Each TRF combines the time-lagged coefficients for one regressor. The resulting time course describes how the feature of interest modulates

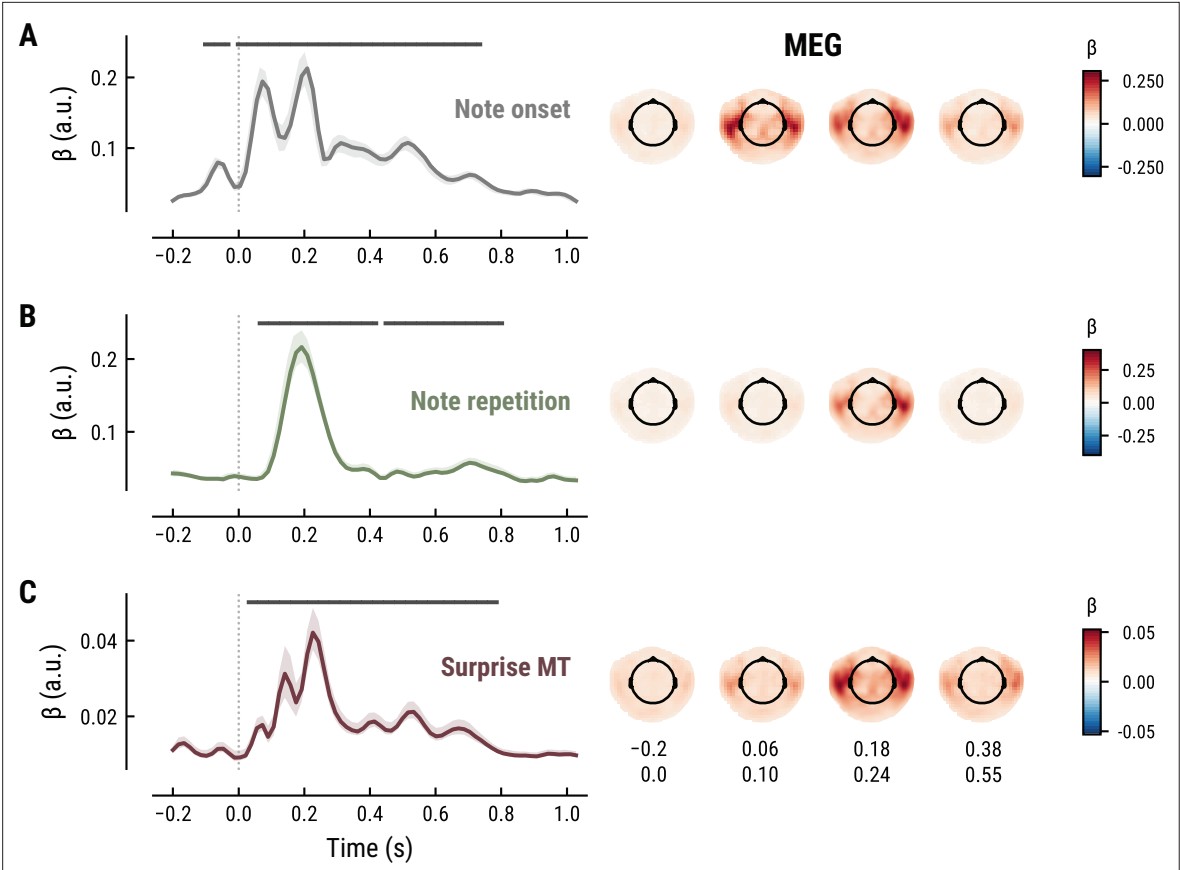

**Figure 5.** Temporal response functions (TRFs, left column) and spatial topographies at four time periods (right column) for the best model on the MEG data. (**A**): Note onset regressor. (**B**): Note repetition regressor. (**C**): Surprise regressor from the Music Transformer with a context length of eight notes. TRF plots: Grey horizontal bars: time points at which at least one channel in the ROI was significant. Lines: mean across participants and channels. Shaded area: 95% CI across participants.

neural activity over time. Here, we focused on note onset, the repetition of notes, and melodic surprise. The TRFs were roughly constant around zero in the baseline period (−0.2–0 s before note onset) and showed a clear modulation time-locked to note onset (*Figures 5 and 6*). This confirmed that the deconvolution of different features and the temporal alignment in the time-resolved regression worked well. Note that the MEG data were transformed to combined planar gradients to yield interpretable topographies (*Bastiaansen and Knösche, 2000*), and therefore did not contain information about the polarity. While we reflect on the sign of modulations in the TRFs below, these judgements were based on inspection of the axial gradiometer MEG results (not shown) and confirmed on the EEG data (*Figure 6*).

The TRF for the note onset regressor reflects the average neural response evoked by a note. The effect was temporally extended from note onset up to 0.8 s (MEG) and 1 s (EEG) and clustered around bilateral fronto-temporal MEG sensors (MEG: cluster-based permutation test p=0.035, *Figure 5A*; EEG: p=5e-04, *Figure 6A*). The time course resembled a P1-N1-P2 complex, typically found in ERP studies on auditory processing (*Picton, 2013*; *Pratt, 2011*), with a first positive peak at about 75ms (P1) and a second positive peak at about 200ms (P2). This was followed by a more sustained negative deflection between 300 and 600ms. We inspected the note repetition regressors to account for the repetition suppression effect, as a potential confound of melodic expectations (*Todorovic et al., 2011*; *Todorovic and de Lange, 2012*). We observed a negative deflection at temporal sensors peaking at about 200ms, reflecting lower neural activity for repeated versus non-repeated notes (MEG: p=5e-04, *Figure 5B*; EEG: p=0.008, *Figure 6B*). This extends the well-known auditory repetition suppression effect (*Grill-Spector et al., 2006*; *Todorovic and de Lange, 2012*) to the setting of naturalistic music listening. Finally, the TRF of the surprise regressor indicates how the level of model-based surprise

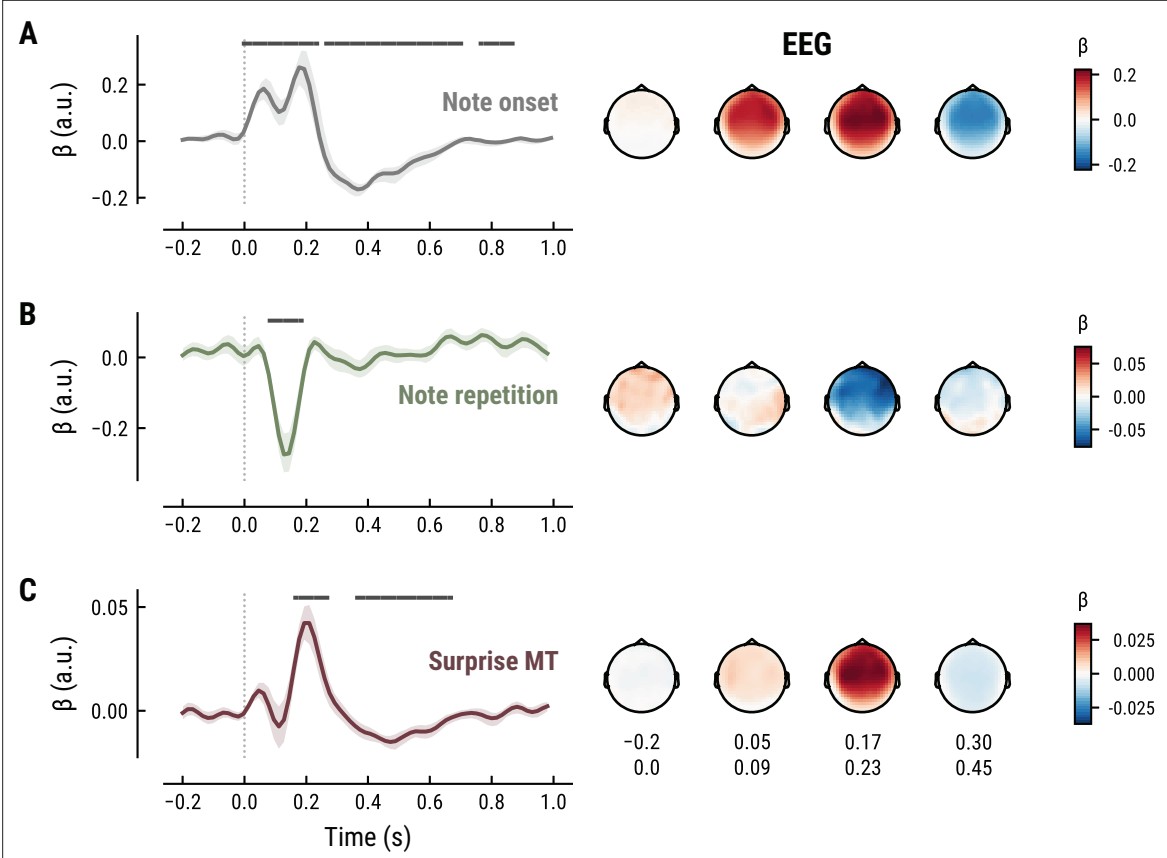

**Figure 6.** All panels as in *Figure 5*, but applied to the EEG data and its musical stimuli.

modulates neural activity over and above simple repetition. A fronto-temporal cluster of MEG sensors exhibited a positive peak at about 200ms and a sustained negative deflection between 300 and 600ms (MEG: p=5e-04, *Figure 5C*; EEG: p=0.004, *Figure 6C*). The increased activity for more surprising notes is consistent with expectation suppression effects (*Todorovic and de Lange, 2012*). We ruled out that the late negativity effect was an artifact arising from a negative correlation between surprise estimates of subsequent notes, since these temporal autocorrelations were consistently found to be positive. The surprise estimates from the Temperley and IDyOM models yielded similar, although slightly weaker, spatiotemporal patterns in the MEG and EEG data (*Appendix 1—figures 3 and 4*), indicating that they all captured melodic surprise given the cross-model correlations.

## Melodic processing is associated with superior temporal and Heschl's gyri

To further shed light on the spatial profile of melody and surprise processing, we estimated the dominant neural sources corresponding to the peak TRF deflection (180–240ms post note onset) using equivalent current dipole (ECD) modeling of the MEG data (with one, two, or three dipoles per hemisphere, selected by comparing adjusted $r^2$). These simple models provided a good fit to the sensor-level TRF maps, indicated by the substantial amount of variance explained (mean adjusted $r^2$ across participants = 0.98 / 0.98/0.97 for Onset / Repetition / Surprise regressors, SD = 0.013 / 0.011/0.020). We show the density of fit dipole locations in *Figure 7*. The TRF peak deflection for the Onset regressor was best explained by sources in bilateral Heschl's gyri (*Figure 7*, top). The peak deflections for the Repetition and Surprise regressors were best explained by slightly more lateral sources encompassing both bilateral Heschl's gyri as well as bilateral superior temporal gyri (see *Figure 7* for exact MNI coordinates of density peaks).

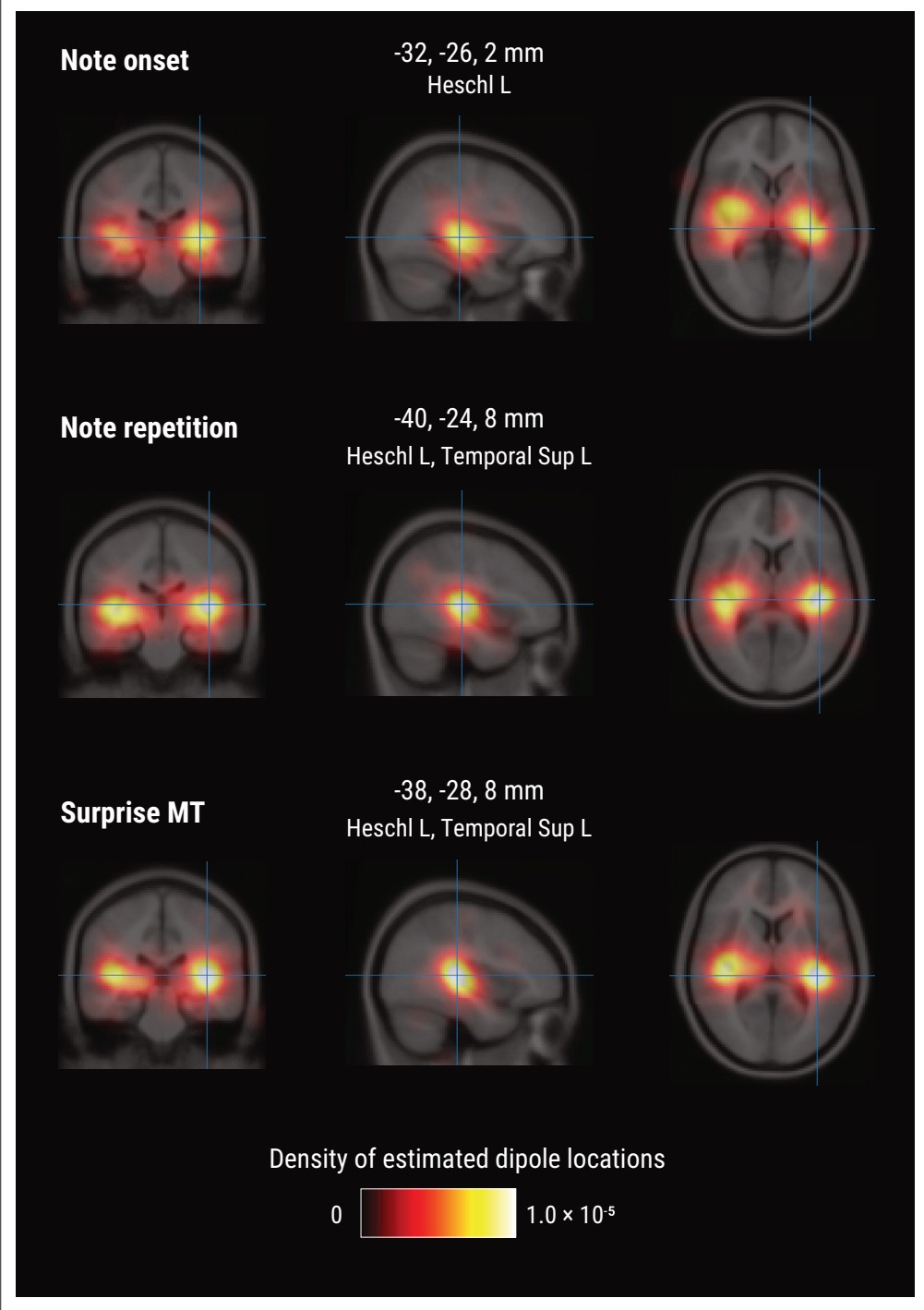

**Figure 7.** Source-level results for the MEG TRF data. Volumetric density of estimated dipole locations across participants in the time window of interest identified in *Figure 5* (180–240ms), projected on the average Montreal Neurological Institute (MNI) template brain. MNI coordinates are given for the density maxima with anatomical labels from the Automated Anatomical Labeling atlas.

## No evidence for neural tracking of melodic uncertainty

Besides surprise, melodic expectations can be characterized by their note-level uncertainty. Estimates of surprise and uncertainty were positively correlated across different computational models (e.g. MT with a context of eight notes: $r=0.21$) (*Figure 8A*). Surprisingly, the addition of uncertainty and its interaction with surprise did not further improve but rather reduce models' cross-validated predictive

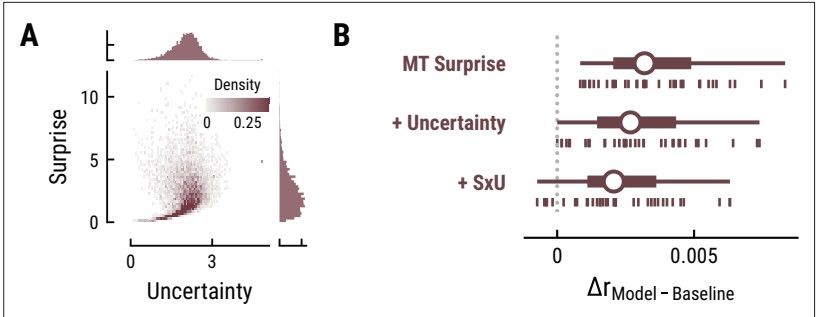

**Figure 8.** Results for melodic uncertainty. (**A**) Relationship between and distribution of surprise and uncertainty estimates from the Music Transformer (context length of eight notes). (**B**) Cross-validated predictive performance for the Baseline +surprise model (top), and for models with added uncertainty regressor (middle) and the interaction between surprise and uncertainty (SxU, bottom). Adding uncertainty and/or the interaction between surprise and uncertainty (SxU) did not improve but worsen the predictive performance on the MEG data.

performance on listeners' MEG data compared to surprise alone (MT Surprise: $\Delta r_{\text{Surpise-Baseline}}$=0.004, SD = 0.002;+Uncertainty: $\Delta r_{\text{Uncertainty-Baseline}}$=0.003, SD = 0.002, paired-sample t-test compared to Surprise, $t_{34}$=−9.57, p=1.42e-10, d=−1.64;+Interaction S×U: $\Delta r_{\text{SxU-Baseline}}$=0.002, SD = 0.002, $t_{34}$=−13.81, p=1.66e-14, d=−2.37) (*Figure 8B*). This result holds true for other computational models of music and for the EEG data. Therefore, we do not further examine the TRFs here.

## Discussion

In the present study, we investigated the nature of melodic expectations during naturalistic music listening. We used a range of computational models to calculate melodic surprise and uncertainty under different internal models. Through time-resolved regression on human listeners' M|EEG activity, we gauged which model could most accurately predict neural indices of melodic surprise. In general, melodic surprise enhanced neural responses, particularly around 200ms and between 300 and 500ms after note onset. This was dissociated from sensory-acoustic and repetition suppression effects, supporting expectation-based models of music perception. In a comparison between computational models of musical expectation, melodic surprise estimates that were generated by an internal model that used *long-term* statistical learning best captured neural surprise responses, highlighting extensive experience with music as a key source of melodic expectations. Strikingly, this effect appeared to be driven by *short-range* musical context of up to 10 notes instead of longer range structure. This provides an important window into the nature and content of melodic expectations during naturalistic music listening.

Expectations are widely considered a hallmark of music listening (*Huron, 2006*; *Koelsch et al., 2019*; *Krumhansl, 2015*; *Meyer, 1957*; *Tillmann et al., 2014*; *Vuust et al., 2022*), which resonates with the predictive coding framework of perception and cognition (*Clark, 2013*; *de Lange et al., 2018*; *Friston, 2010*). Here, we tested the role of melodic expectations during naturalistic music listening, for which neural evidence has been scarce. We quantified note-level surprise and uncertainty as markers of melodic expectations and examined their effect on neural music processing using time-resolved regression. Importantly, our analyses focused on disentangling different sources of melodic expectations, as well as elucidating the length of temporal context that the brain is taking into account when predicting which note will follow. This represents a critical innovation over earlier related work (*Di Liberto et al., 2020*), from which conclusions were necessarily limited to establishing *that* the brain predicts something during music listening, whereas we begin to unravel *what* it is that is being predicted. Furthermore, our use of diverse naturalistic musical stimuli and MEG allows for a broader generalization of our conclusions than was previously possible. Of course, the stimuli do not fully reflect the richness of real-world music yet, as for example the MIDI velocity (i.e. loudness) was held constant and only monophonic compositions were presented. Monophony was a technical limitation given the application of the Temperley and IDyOM model. The reported performance of the MusicTransformer, which supports fully polyphonic music, opens new avenues for future work studying the neural basis of music processing in settings even closer to fully naturalistic.

A key signature of predictive auditory processing is the neural response to unexpected events, also called the prediction error response (*Clark, 2013*; *Friston, 2010*; *Heilbron and Chait, 2018*). The degree to which notes violate melodic expectations can be quantified as the melodic surprise. Across different computational models of music, we found that melodic surprise explained M|EEG data from human listeners beyond sensory-acoustic factors and beyond simple repetition effects. We thereby generalize previous behavioral and neural evidence for listeners' sensitivity to unexpected notes to a naturalistic setting (for reviews see *Koelsch et al., 2019*; *Rohrmeier and Koelsch, 2012*; *Tillmann et al., 2014*; *Zatorre and Salimpoor, 2013*).

While the role of expectations in music processing is well established, there is an ongoing debate about the *nature* of these musical expectations (*Bigand et al., 2014*; *Collins et al., 2014*; *Rohrmeier and Koelsch, 2012*). It has been claimed that these stem from a small set of general, Gestalt-like, principles (*Krumhansl, 2015*; *Temperley, 2008*; *Temperley, 2014*). Alternatively, they may reflect the outcome of a statistical learning process (*Pearce, 2005*; *Pearce and Wiggins, 2012*; *Rohrmeier and Koelsch, 2012*), which, in turn, could reflect either short- or long-range regularities. For the first time, we present neural evidence that weighs in on these questions. We simulated note-level expectations from different predictive architectures of music, which reflected distinct sources of melodic expectations: Gestalt-like principles (Temperley model), short-term statistical learning during the present composition (IDyOM stm) or statistical learning through long-term exposure to music (IDyOM ltm, Music Transformer).

As a first core result, we found that long-term statistical learning (Music Transformer and IDyOM ltm) captured neural surprise processing better than short-term regularities or Gestalt principles. Our results thus stress the role of long-term exposure to music as a central source of neural melodic expectations. The human auditory system exhibits a remarkable sensitivity to detect and learn statistical regularities in sound (*Saffran et al., 1999*; *Skerritt-Davis and Elhilali, 2018*). This capacity has been corroborated in statistical learning paradigms using behavioral (*Barascud et al., 2016*; *Bianco et al., 2020*), eye-tracking (*Milne et al., 2021*; *Zhao et al., 2019*), and neuroimaging techniques (*Barascud et al., 2016*; *Moldwin et al., 2017*; *Pesnot Lerousseau and Schön, 2021*). Furthermore, humans have extraordinary implicit memory for auditory patterns (*Agres et al., 2018*; *Bianco et al., 2020*). It has therefore been proposed that listeners learn the statistical regularities embedded in music through mere exposure (*Pearce, 2018*; *Rohrmeier et al., 2011*; *Rohrmeier and Rebuschat, 2012*).

Short-term regularities and Gestalt principles also significantly predicted neural variance and might constitute concurrent, though weaker, sources of melodic expectations (*Rohrmeier and Koelsch, 2012*). Gestalt principles, specifically, have been shown to adequately model listeners' melodic expectations in behavioral studies (*Cuddy and Lunney, 1995*; *Morgan et al., 2019*; *Pearce and Wiggins, 2006*; *Temperley, 2014*). One shortcoming of Gestalt-like models, however, is that they leave unresolved how Gestalt rules emerge, assuming either innate principles (*Narmour, 1990*) or being agnostic to this question (*Temperley, 2008*). We propose that the well-established statistical learning framework can account for Gestalt-like principles. If the latter, for example pitch proximity, indeed fit a certain musical style, they have to be reflected in the statistical regularities. Music theoretical research has indeed shown that statistical learning based on bigrams can recover music theoretical Gestalt principles (*Rodriguez Zivic et al., 2013*), even across different (musical) cultures (*Savage et al., 2015*). This further backs up the role of statistical learning for musical expectations.

As a second core result, strikingly, we found that neural activity was best explained by those surprise estimates taking into account only relatively short-range musical context. Even though extracting the patterns upon which expectations are based requires long-term exposure (previous paragraph), the relevant context length of these patterns for predicting upcoming notes turned out to be short, around 7–8 notes. In contrast, for modeling music itself (i.e. independently of neural activity), the music transformer performed monotonically better with increasing context length, up to hundreds of notes. This pattern of results is very unlike similar studies in language processing, where models that perform best at next word prediction and can take the most context into account (i.e. transformers) also perform best at predicting behavioral and brain responses, and predictions demonstrably take long-term context into account (*Goodkind and Bicknell, 2018*; *Heilbron et al., 2021*; *Schmitt et al., 2021*; *Schrimpf et al., 2021*; *Wilcox et al., 2020*). A cautious hypothesis is that musical motifs, groups of about 2–10 notes, are highly generalizable within a musical style compared to longer range structure (*Krumhansl, 2015*). Motifs might thus drive statistical learning and melodic predictions,

while other temporal scales contribute concurrently (*Maheu et al., 2019*). However, several alternative explanations are possible, between which we cannot adjudicate, based on our data. First, the length of ten notes roughly corresponds to the limit of auditory short-term memory at about 2–4 s (*Thaut, 2014*), which might constrain predictive sequence processing. Second, our analysis is only sensitive to time-locked note-level responses and those signals measured by M|EEG, whereas long-range musical structure might have different effects on neural processing (*Krumhansl, 2015*; *Rohrmeier and Koelsch, 2012*), in particular slower effects that are less precisely linked to note onsets. A third and final caveat is that the modeling of long-range structure by the music transformer model might be different from how human listeners process temporally extended or hierarchical structure.

Our approach of using temporal response function (TRF, or 'regression evoked response', rERP) analysis allowed us to investigate the spatiotemporal characteristics of continuously unfolding neural surprise processing. Melodic surprise modulated neural activity evoked by notes over fronto-temporal sensors with a positive peak at about 200ms, corresponding to a modulation of the P2 component (*Picton, 2013*; *Pratt, 2011*). Source modeling suggests superior temporal and Heschl's gyri as likely sources of this neural response (although we note that MEG's spatial resolution is limited and the exact localization of surprise responses within auditory cortex requires further research). Surprising notes elicited stronger neural responses, in line with previous reports by *Di Liberto et al., 2020*. This finding is furthermore consistent with the more general effect of expectation suppression, the phenomenon that expected stimuli evoke weaker neural responses (*Auksztulewicz and Friston, 2016*; *Garrido et al., 2009*; *Todorovic and de Lange, 2012*; *Wacongne et al., 2011*) through gain modulation (*Quiroga-Martinez et al., 2021*). In line with predictive coding, the brain might hence be predicting upcoming notes in order to explain away predicted sensory input, thereby leading to enhanced responses to surprising (i.e., not yet fully explainable) input.

Additionally, we found a sustained late negativity correlating with melodic surprise, which some studies have labeled a musical N400 or N500 (*Calma-Roddin and Drury, 2020*; *Koelsch et al., 2000*; *Miranda and Ullman, 2007*; *Painter and Koelsch, 2011*; *Pearce et al., 2010*). Similar to its linguistic counterpart (*Kutas and Federmeier, 2011*), the N400 has been interpreted as an index of predictive music processing. The literature has furthermore frequently emphasised the mismatch negativity (MMN) (*Näätänen et al., 2007*) and P3 component in predictive music processing (*Koelsch et al., 2019*), neither of which we observe for melodic surprise here. However, the MMN is typically found for deviants occurring in a stream of standard tones, such as in oddball paradigms, while the P3 is usually observed in the context of an explicit behavioral task (*Koelsch et al., 2019*). In our study, listeners were listening passively to maximize the naturalistic setting, which could account for the absence of these components. Importantly, our results go beyond previous research by analysing the influence of melodic surprise in a continuous fashion, instead of focusing on deviants.

As a final novel contribution, we demonstrate the usefulness of a state-of-the-art deep learning model, the Music Transformer (MT) (*Huang et al., 2018*), for the study of music cognition. The network predicted music and neural data at least on par with the IDyOM model, an n-gram model which is currently a highly popular model of musical expectations (*Pearce and Wiggins, 2012*). We are likely severely underestimating the relative predictive power of the MT, since we constrained our stimuli to monophonic music in the present study. Monophonic music is the only type of music the other models (IDyOM, Temperley) are able to process, so this restriction was a technical necessity. The MT, in contrast, supports fully polyphonic music. This opens up new avenues for future work to study neural music processing in even more naturalistic settings.

To conclude, by using computational models to capture different hypotheses about the nature and source of melodic expectations and linking these to neural data recorded during naturalistic listening, we found that these expectations have their origin in long-term exposure to the statistical structure of music. Yet, strikingly, as listeners continuously exploit this long-term knowledge during listening, they do so primarily on the basis of short-range context. Our findings thereby elucidate the individual voices making up the 'surprise symphony' of music perception.

# Materials and methods

## Data and code availability

The (anonymized, de-identified) MEG and music data are available from the Donders Repository (https://data.donders.ru.nl/) under CC-BY-4.0 license. The persistent identifier for the data is https://doi.org/10.34973/5qxw-nn97. The experiment and analysis code is also available from the Donders Repository.

## Participants

We recruited 35 healthy participants (19 female; 32 right-handed; age: 18–30 years, mean = 23.8, SD = 3.05) via the research participation system at Radboud University. The sample size was chosen to achieve a power of ≥80% for detecting a medium effect size (d=0.5) with a two-sided paired t-test at an α level of 0.05. All participants reported normal hearing. The study was approved under the general ethical approval for the Donders Centre for Cognitive Neuroimaging (Imaging Human Cognition, CMO2014/288) by the local ethics committee (CMO Arnhem-Nijmegen, Radboud University Medical Centre). Participants provided written informed consent before the experiment and received monetary compensation.

## Procedure

Participants listened to music, while their neural activity was recorded using magnetoencephalography (MEG) (*Figure 1*). Participants started each musical stimulus with a button press and could take short breaks in between stimuli. Participants were instructed to fixate a dot displayed at the centre of a screen (~85 cm viewing distance) in order to reduce head and eye movements. Besides that, participants were only asked to listen attentively to the music and remain still. These minimal instructions were intended to maximize the naturalistic character of the study. Initially, three test runs (~10 s each) were completed, in which three short audio snippets from different compositions (not used in the main experiment) were presented. This was intended to make listeners familiar with the procedure and the different sounds, as well as to adjust the volume to a comfortable level.

## Musical stimuli

We selected 19 compositions (duration: total = 43 min, median across stimuli = 134 s, median absolute deviation (MAD, *Leys et al., 2013*) = 39 s; note events: total = 9824, median = 448, MAD = 204) from Western classical music (see *Appendix 1—table 1*). We chose this genre, since (a) participants recruited from the Nijmegen area were assumed to be somewhat familiar with it, (b) it entails relatively complex melodies and long-term structure allowing us to sample a broad range of surprise and uncertainty estimates, (c) many digital music files and corpora in MIDI format are publicly available, and (d) these included monophonic pieces. Monophonic refers to one note being played at a time, that is only containing a melody, compared to polyphonic music, which further includes chords and/or parallel voices. The constraint to monophonic compositions was necessary to enable the application of the Temperley and IDyOM model, which cannot parse polyphonic music. Based on the available databases, the selection aimed to cover various musical periods (1711–1951), composers, tempi (60–176 bpm), and key signatures, roughly matching the statistics of the training corpus for the music models (see below). The median note duration was about 161ms (MAD across all notes = 35ms, min = 20ms, max = 4498ms), with a median inter-note onset interval of 200ms (MAD across all notes = 50ms, min = 22ms, max = 2550ms).

We used the Musescore 3 software to synthesize and export the digital MIDI files as wav audio files (sampling rate = 44.1 kHz). This ensured accurate control over the note timing compared to live or studio recordings, facilitating time-locked analyses. The synthesisation via one of three virtual instruments from fluidsynth (piano, oboe, flute) ensured the natural character of the music. The MIDI velocity, corresponding to loudness (termed 'velocity' in MIDI terms because it refers to the velocity with which one could strike a piano key), was set to 100 for all notes, since most files were missing velocity information and the volume was thus held roughly constant across notes.

## Stimulus presentation

The experiment was run on a Windows computer using Matlab 2018b (The MathWorks) and the Psychophysics Toolbox (*Brainard, 1997*). The music was presented binaurally via ear tubes (Doc's

Promolds NonVent with #13 thick prebent 1.85 mm ID tubes, Audine Healthcare, in combination with Etymotic ER3A earphones) at a sampling rate of 44.1 kHz. The volume was adjusted to a comfortable level for each participant during the initial three test runs. To ensure equivalent acoustic input in both ears, the right audio channel from potentially stereo recordings was duplicated, resulting in mono audio presentation. After participants initiated a run by a button press, the wav file was first loaded into the sound card buffer to ensure accurate timing. Once the file was fully loaded, the visual fixation cross appeared at the centre of the screen and after 1.5–2.5 s (random uniform distribution) the music started. The order of compositions was randomized across participants.

## MEG data acquisition

Neural activity was recorded on a 275-channel axial gradiometer MEG system (VSM/CTF Systems) in a magnetically shielded room, while the participant was seated. Eight malfunctioning channels were disabled during the recording or removed during preprocessing, leaving 267 MEG channels in the recorded data. We monitored the head position via three fiducial coils (left and right ear, nasion). When the head movement exceeded 5 mm, in between listening periods, the head position was shown to the participant, and they were instructed to reposition themselves (*Stolk et al., 2013*). All data were low-pass filtered online at 300 Hz and digitized at a sampling rate of 1200 Hz.

## Further data acquisition

For source analysis, the head shape and the location of the three fiducial coils were measured using a Polhemus 3D tracking device. T1-weighted anatomical MRI scans were acquired on a 3T MRI system (Siemens) after the MEG session if these were not already available from the local database (MP-RAGE sequence with a GRAPPA acceleration factor of 2, TR = 2.3 s, TE = 3.03ms, voxel size 1 mm isotropic, 192 transversal slices, 8 ° flip angle). Additionally, during the MEG session, eye position, pupil diameter and blinks were recorded using an Eyelink 1000 eye tracker (SR Research) and digitized at a sampling rate of 1200 Hz. After the experiment, participants completed a questionnaire including a validated measure of musicality, the Goldsmith Musical Sophistication Index (*Müllensiefen et al., 2014*). The eye tracking and questionnaire data were not analysed here.

## EEG dataset

In addition, we analysed an open data set from a recently published study (*Di Liberto et al., 2020*) including EEG recordings from 20 participants (10 musicians, 10 non-musicians) listening to music. The musical stimuli were 10 violin compositions by J. S. Bach synthesized using a piano sound (duration: total = 27 min, median = 161.5 s, MAD = 18.5 s; note events: total = 7839,, median = 631, MAD = 276.5; see *Appendix 1—table 1*), that were each presented three times in pseudo-randomized order (total listening time = 80 min). The median note duration was 145ms (MAD across all notes = 32ms, min = 70ms, max = 2571ms), with a median inter-note onset interval of 150ms (MAD across all notes = 30ms, min = 74ms, max = 2571ms). EEG was acquired using a 64-electrode BioSemi Active Two system and digitized at a sampling rate of 512 Hz.

## Music analysis

We used three types of computational models of music to investigate human listeners' melodic expectations: the Temperley model (*Temperley, 2008*; *Temperley, 2014*), the IDyOM model (*Pearce and Wiggins, 2012*), and the Music Transformer (*Huang et al., 2018*). Based on their differences in computational architecture, we used these models to operationalize different sources of melodic expectations. All models take as input MIDI data, specifically note pitch values X ranging discretely from 0 to 127 (8.18–12543.85 Hz, middle C=60,~264 Hz). The models output a probability distribution for the next note pitch at time point $t$, $X_t$, given a musical context of $k$ preceding consecutive note pitches:

$$P(X_t \mid x_{t-k}^{t-1}), \text{ where } X \in \{0..127\}, k > 0, t \geq 0.$$

For the first note in each composition, we assumed a uniform distribution across pitches ($P(X_0 = x) = 1/128$). Based on these probability distributions, we computed the surprise $S$ of an observed note pitch $x_t$ given the musical context as

$$S\left(x_t\right) = -\log_e P\left(x_t | x_{t-k}^{t-1}\right).$$

Likewise, the uncertainty $U$ associated with predicting the next note pitch was defined as the entropy of the probability distribution across all notes in the alphabet:

$$U_t = -\sum_{x=0}^{127} P\left(X_t = x | x_{t-k}^{t-1}\right) \log_e P\left(X_t = x | x_{t-k}^{t-1}\right).$$

### Training corpora

All models were trained on the Monophonic Corpus of Complete Compositions (MCCC) (https://osf.io/dg7ms/), which consists of 623 monophonic pieces (Note events: total = 500,000, median = 654, MAD = 309). The corpus spans multiple musical periods and composers and matches the statistics of the musical stimuli used in the MEG and EEG study regarding the distribution of note pitch and pitch interval (*Appendix 1—figure 1*) as well as the proportion of major key pieces (MCCC: ~81%, Music$_{MEG}$: ~74%, but Music$_{EEG}$: 20%). Furthermore, the Maestro corpus V3 (*Hawthorne et al., 2019*, https://magenta.tensorflow.org/datasets/maestro), which comprises 1276 polyphonic compositions collected from human piano performances (Duration: total = 200 h, note events: total = 7 million), was used for the initial training of the Music Transformer (see below).

### Probabilistic Model of Melody Perception | Temperley

The Probabilistic Model of Melody Perception (*Temperley, 2008*; *Temperley, 2014*) is a Bayesian model based on three interpretable principles established in musicology. Therefore, it has been coined a Gestalt-model (*Morgan et al., 2019*). The three principles are modeled by probability distributions (discretized for integer pitch values), whose free parameters were estimated, in line with previous literature, based on the MCCC:

1. Pitches $x_t$ cluster in a narrow range around a central pitch $c$ (central pitch tendency):

$$x_t \sim \mathcal{N}(c, v_r), \text{where } c \sim \mathcal{N}\left(c_0, var_{c0}\right).$$

   The parameters $c_0$ and $var_{c0}$: were set to the mean and variance of compositions' mean pitch in the training corpus ($c_0$=72, $var_{c0}$ = 34.4). The variance of the central pitch profile $v_r$ was set to the variance of each melody's first note around its mean ($v_r$ = 83.2).

2. Pitches tend to be close to the previous pitch $x_{t-1}$, in other words pitch intervals tend to be small (pitch proximity):

$$x_t \sim \mathcal{N}\left(x_{t-1}, v_x\right)$$

   The variance of the pitch proximity profile $v_x$ was estimated as the variance of pitches around $x_{t-1}$ considering only notes where $x_{t-1}=c$ ($v_x$ = 18.2).

3. Depending on the key, certain pitches occur more frequently given their scale degree (the position of a pitch relative to the tonic of the key). This key profile is modeled as the probability of a scale degree conditioned on the key (12 major and 12 minor keys) spread out across several octaves, weighted by the probability of major and minor keys ($p_{maj}$ = .81).

The final model multiplicatively combines these distributions to give the probability of the next note pitch given the context. The C code was provided by David Temperley in personal communication and adapted to output probabilities for all possible pitch values $X$. Specific choices in principles 1–3 above were made in accordance with earlier work (*Morgan et al., 2019*; *Temperley, 2008*; *Temperley, 2014*).

### Information Dynamics of Music model | IDyOM

The Information Dynamics of Music (IDyOM) model is an unsupervised statistical learning model, specifically a variable order Markov model (*Pearce, 2005*; *Pearce and Wiggins, 2012*). Based on n-grams and the alphabet $X$, the probability of a note pitch $x$ at time point $t$, $x_t$, given a context sequence of length $k$, $x_{t-k}^{t-1}$, is defined as the relative n-gram frequency of the continuation compared to the context:

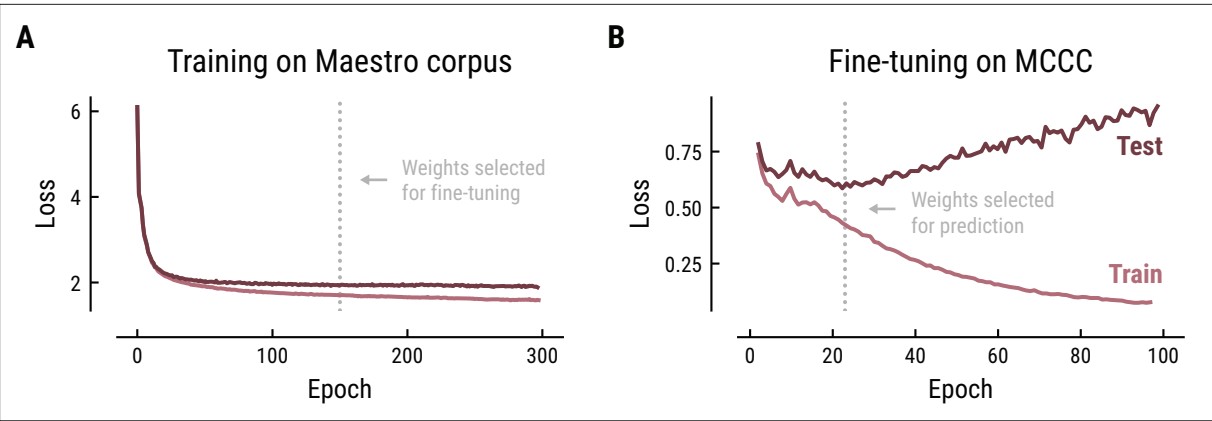

**Figure 9.** Training (**A**) and fine-tuning (**B**) of the Music Transformer on the Maestro corpus and MCCC, respectively. Cross-entropy loss (average surprise across all notes) on the test (dark) and training (light) data as a function of training epoch.

$$P(x_t \mid x_{t-k}^{t-1}) = \frac{count(x_{t-k}^t)}{count(x_{t-k}^{t-1})}.$$

The probabilities are computed for every possible n-gram length up to a bound *k* and combined through interpolated smoothing. The context length was, therefore, manipulated via the n-gram order bound. The model can operate on multiple musical features, called viewpoints. Here, we use pitch (in IDyOM terminology *cpitch*) to predict pitch, in line with the other models.

The IDyOM model class entails three different subtypes: a short-term model (stm), a long-term model (ltm), and a combination of the former two (both). The IDyOM stm model rests solely on the recent context in the current composition. As such, it approximates online statistical learning of short-term regularities in the present piece. The IDyOM ltm model, on the other hand, is trained on a corpus, reflecting musical enculturation, that is (implicit) statistical learning through long-term exposure to music. The IDyOM both model combines the stm and ltm model weighted by their entropy at each note.

## Music Transformer

The Music Transformer (MT) (*Huang et al., 2018*) is a state-of-the-art neural network model that was developed to generate music with improved long-range coherence. To this end, it takes advantage of a Transformer architecture (*Vaswani et al., 2017*) and relative self-attention (*Shaw et al., 2018*), which better capture long-range structure in sequences than for example n-gram models. The MT is the only model used here that can process polyphonic music. This is possible due to a representation scheme that comprises four event types (note onset, note offset, velocity, and time-shift events) for encoding and decoding MIDI data. The note onset values are equivalent to pitch values and were used to derive probability distributions. Our custom scripts were based on an open adaptation for PyTorch (https://github.com/gwinndr/MusicTransformer-Pytorch; *Gwinn et al., 2022*).

The Music Transformer was initially trained on the polyphonic Maestro corpus for 300 epochs using the training parameters from the original paper (learning rate = 0.1, batch size = 2, number of layers = 6, number of attention heads = 6, dropout rate = 0.1, *Huang et al., 2018*). The training progress was monitored based on the cross-entropy loss on the training data (80%) and test data (20%) (*Figure 9A*). The cross-entropy loss is defined as the average surprise across all notes. The model is, thus, trained to minimize the surprise for upcoming notes. The minimal loss we achieved (1.97) was comparable to the original paper (1.835). The divergence between the loss curve for training and test set indicated some overfitting starting from about epoch 50, however, without a noticeable decrease in test performance. Therefore, we selected the weights at epoch 150 to ensure stable weights without severe overfitting.

In order to adjust the model to monophonic music, we finetuned the pretrained Music Transformer on the MCCC for 100 epochs using the same training parameters (*Figure 9B*). Again, the training progress was evaluated based on the cross-entropy loss and the weights were selected based on the minimal loss. While the loss started at a considerably lower level on this monophonic dataset (0.78), it

continued to decrease until epoch 21 (0.59), but quickly started to increase, indicating overfitting on the training data. Therefore, the weights from epoch 21 were selected for further analyses.

## Music model comparison

We compared the models' predictive performance on music data as a function of model class and context length. Thereby, we aimed to scrutinize the hypothesis that the models reflect different sources of melodic expectations. We used the musical stimuli from the MEG and EEG study as test sets and assessed the accuracy, median surprise and uncertainty across compositions.

## M|EEG analysis

### Preprocessing

The MEG data were preprocessed in Matlab 2018b using FieldTrip (*Oostenveld et al., 2011*). We loaded the raw data separately for each composition including about 3 s pre- and post-stimulus periods. Based on the reference sensors of the CTF MEG system, we denoised the recorded MEG data using third-order gradient correction, after which the per-channel mean across time was subtracted. We then segmented the continuous data in 1 s segments. Using the semi-automatic routines in FieldTrip, we marked noisy segments according to outlying variance, such as MEG SQUID jumps, eye blinks or eye movements (based on the unfiltered data) or muscle artifacts (based on the data filtered between 110 and 130 Hz). After removal of noisy segments, the data were downsampled to 400 Hz. Independent component analysis (ICA) was then performed on the combined data from all compositions for each participant to identify components that reflected artifacts from cardiac activity, residual eye movements or blinks. Finally, we reloaded the data without segmentation, removed bad ICA components and downsampled the data to 60 Hz for subsequent analyses.

A similar preprocessing pipeline was used for the EEG data. Here, the data were re-referenced using the linked mastoids. Bad channels were identified via visual inspection and replaced through interpolation after removal of bad ICA components.

### TRF analysis

We performed time-resolved linear regression on the M|EEG data to investigate the neural signatures of melodic surprise and uncertainty (*Figure 1*), using the regression evoked response technique ('rERP', *Smith and Kutas, 2015*).This approach allowed us to deconvolve the responses to different features and subsequent notes and correct for their temporal overlap. The preprocessed M|EEG data were loaded and band-pass filtered between 0.5 and 8 Hz (bidirectional FIR filter). All features of interest were modeled as impulse regressors with one value per note, either binary (x = {0,1}) or continuous ($x \in R$). The M|EEG channel data and continuous regressors were z-scored. We constructed a time-expanded regression matrix $M$, which contained time-shifted versions of each regressor column-wise ($t_{min}$ = –0.2 s, $t_{max}$ = 1 s relative to note onsets, 73 columns per regressor given the sampling rate of 60 Hz). After removal of bad time points identified during M|EEG preprocessing, we estimated the regression weights $\hat{\beta}$ using ordinary least squares (OLS) regression:

$$\hat{\beta} = \left(M^T M\right)^{-1} M^T y.$$

Collectively, the weights form a response function known as the regression evoked response or temporal response function (TRF; *Crosse et al., 2016*; *Ding and Simon, 2012*). The TRF depicts how a feature modulates neural activity across time. Here, the units are arbitrary, since both binary and z-scored continuous regressors were included. Model estimation was performed using custom Python code built on the MNE rERP implementation (*Gramfort et al., 2013*; *Smith and Kutas, 2015*). Previous similar work has used ridge-regularized regression, rather than OLS (*Di Liberto et al., 2020*). We instead opted to use OLS, since the risk for overfitting was low given the sparse design matrices and low correlations between the time-shifted regressors. To make sure this did not unduly influence our results, we also implemented ridge-regularized regression with the optimal cost hyperparameter alpha estimated via nested cross-validation. OLS (alpha = 0) was always among the best-fitting models and any increase in predictive performance for alpha >0 for some participants was negligible. Results for this control analysis are shown for the best fitting model for the MEG and EEG data in

*Appendix 1—figures 5 and 6*, respectively. In the rest of the manuscript we thus report the results from the OLS regression.

## Models and regressors

The **Onset model** contained a binary regressor, which coded for note onsets and was included in all other models too. The **Baseline model** added a set of regressors to control for acoustic properties of the music and other potential confounds. Binary regressors were added to code for (1) very high pitch notes (>90% quantile), (2) very low pitch notes (<10% quantile), since extreme pitch values go along with differences in perceived loudness, timbre, and other acoustic features; (3) the first note in each composition (i.e. composition onset); (4) repeated notes, to account for the repetition suppression effect and separate it from the surprise response. Since the MEG experiment used stimuli generated by different musical instruments, we additionally controlled for the type of sound, by including binary regressors for oboe and flute sounds. This was done since the different sounds have different acoustic properties, such as a lower attack time for piano sounds and longer sustain for oboe or flute sounds. For computing continuous acoustic regressors, we downsampled the audio signal to 22.05 kHz. We computed the mean for each variable of interest across the note duration to derive a single value for each note and create impulse regressors. The root-mean-square value (RMS) of the audio signal captures differences in (perceived) loudness. Flatness, defined as the ratio between the geometric and the arithmetic mean of the acoustic signal, controlled for differences in timbre. The variance of the broad-band envelope represented acoustic edges (*McDermott and Simoncelli, 2011*). The broad-band envelope was derived by (a) filtering the downsampled audio signal through a gammatone filter bank (64 logarithmically spaced filter bands ranging between 50 and 8000 Hz), which simulates human auditory processing; (b) taking the absolute value of the Hilbert transform of the 64 band signals; (c) averaging across bands (*Zuk et al., 2021*). The baseline regressors were also included in all of the following models. The **main models** of interest added note-level surprise, uncertainty, and/or their interaction from the different computational models of music, varying the model class and context length.

## Model comparison

We applied a fivefold cross-validation scheme (train: 80%, test: 20%, time window: 0–0.6 s) (*Varoquaux et al., 2017*) to compare the regression models' predictive performance on the M|EEG data. We computed the correlation between the predicted and recorded neural signal across time for each fold and channel on the hold out data. To increase the sensitivity of subsequent analyses, we selected the channels most responsive to musical notes for each participant according to the cross-validated performance for the Onset model (>2/3 quantile). The threshold was determined through visual inspection of the spatial topographies, but did not affect the main results. The overall model performance was then determined as the median across folds and the mean across selected channels. Since the predictive performance was assessed on unseen hold out data, the approach controlled for overfitting the neural data and for differences in the number of regressors and free model parameters. For statistical inference, we computed one-sample or paired t-tests using multiple comparison correction (Bonferroni-Holm method).

## Cluster-based statistics

For visualizations and cluster-based statistics, we transformed the regression coefficients from the axial MEG data to a planar representation using FieldTrip (*Bastiaansen and Knösche, 2000*). The regression coefficients estimated on the axial gradient data were linearly transformed to planar gradient data, for which the resulting synthetic horizontal and vertical planar gradient components were then non-linearly combined to a single magnitude per original MEG sensor. For the planar-transformed coefficients, we selected the most responsive channels according to the coefficients of the note onset regressor in the Onset model (>5/6 quantile, time window: 0–0.6 s). The threshold was determined through visual inspection of the spatial topographies, but did not affect the main results. We then used cluster-based permutation tests (*Maris and Oostenveld, 2007*) to identify significant spatio-temporally clustered effects compared to the baseline time window (−0.2–0 s, 2000 permutations). Using threshold free cluster enhancement (TFCE, *Smith and Nichols, 2009*), we further determined significant time points, where at least one selected channel showed a significant effect.

Mass-univariate testing was done via one-sample t-tests on the baseline-corrected M|EEG data with 'hat' variance adjustment (σ=1e−3) (*Ridgway et al., 2012*).

## Source analysis

To localize the neural sources associated with the different regressors, we used equivalent current dipole modeling (ECD). Individuals' anatomical MRI scans were realigned to CTF space based on the headshape data and the fiducial coil locations, using a semi-automatic procedure in Fieldtrip. The lead field was computed using a single-shell volume conduction model (*Nolte, 2003*). Based on individuals' time-averaged axial gradient TRF data in the main time window of interest (180–240ms), we used a non-linear fitting algorithm to estimate the dipole configuration that best explained the observed sensor maps (FieldTrip's ft_dipolefitting). We compared three models with one to three dipoles per hemisphere. As the final solution per participant, we chose that with the largest adjusted-$r^2$ score in explaining the observed sensor topography (thereby adjusting for the additional 12 free parameters caused by introducing an extra dipole; 2 hemispheres times x/y/z/dx/dy/dz). As starting point for the search, we roughly specified bilateral primary auditory cortex (MNI coordinates x/y/z [48, -28, 10] mm (R), [-40,–28, 6] mm (L); *Anderson et al., 2011*; *Kiviniemi et al., 2009*), with a small random jitter (normally distributed with SD = 1 mm) to prevent exact overlap in starting positions of multiple dipoles. Note that the initial dipole location has a negligible effect on the final solution if the data are well explained by the final fit model. This was the case for our data, see Results. For visualization, we estimated the (volumetric) density of best-fit dipole locations across participants and projected this onto the average MNI brain template, separately for each regressor.

# Acknowledgements

We thank David Temperley for providing the code for his model and Marcus Pearce for discussions on the IDyOM model. This work was supported by The Netherlands Organisation for Scientific Research (NWO Veni grant 016.Veni.198.065 awarded to ES) and the European Research Council (ERC Consolidator grant SURPRISE # 101000942 awarded to FPdL).

# Additional information

### Competing interests

Floris P de Lange: Senior editor, eLife. The other authors declare that no competing interests exist.

### Funding

| Funder | Grant reference number | Author |
| --- | --- | --- |
| Nederlandse Organisatie voor Wetenschappelijk Onderzoek | 016.Veni.198.065 | Eelke Spaak |
| European Research Council | 101000942 | Floris P de Lange |

The funders had no role in study design, data collection and interpretation, or the decision to submit the work for publication.

### Author contributions

Pius Kern, Conceptualization, Data curation, Formal analysis, Visualization, Methodology, Writing - original draft, Project administration, Writing – review and editing; Micha Heilbron, Conceptualization, Supervision, Investigation, Methodology, Project administration; Floris P de Lange, Conceptualization, Supervision, Funding acquisition, Project administration, Writing – review and editing; Eelke Spaak, Conceptualization, Data curation, Software, Formal analysis, Supervision, Funding acquisition, Investigation, Visualization, Methodology, Project administration, Writing – review and editing

### Author ORCIDs

Pius Kern http://orcid.org/0000-0003-4796-1864

Floris P de Lange  http://orcid.org/0000-0002-6730-1452
Eelke Spaak  http://orcid.org/0000-0002-2018-3364

### Ethics

Human subjects: The study was approved under the general ethical approval for the Donders Centre for Cognitive Neuroimaging (Imaging Human Cognition, CMO2014/288) by the local ethics committee (CMO Arnhem-Nijmegen, Radboud University Medical Centre). Participants provided written informed consent before the experiment and received monetary compensation.

### Decision letter and Author response

Decision letter https://doi.org/10.7554/eLife.80935.sa1
Author response https://doi.org/10.7554/eLife.80935.sa2

## Additional files

### Supplementary files

• MDAR checklist

### Data availability

All data have been deposited into the Donders Repository under CC-BY-4.0 license, under identifier https://doi.org/10.34973/5qxw-nn97.

The following dataset was generated:

| Author(s) | Year | Dataset title | Dataset URL | Database and Identifier |
| --- | --- | --- | --- | --- |
| Kern P, Heilbron M, de Lange FP, Spaak E | 2022 | Tracking predictions in naturalistic music listening using MEG and computational models of music | https://doi.org/10.34973/5qxw-nn97 | Donders Repository, 10.34973/5qxw-nn97 |

The following previously published dataset was used:

| Author(s) | Year | Dataset title | Dataset URL | Database and Identifier |
| --- | --- | --- | --- | --- |
| DiLiberto et al | 2020 | Cortical encoding of melodic expectations in human temporal cortex | https://doi.org/10.5061/dryad.g1jwstqmh | Dryad Digital Repository, 10.5061/dryad.g1jwstqmh |

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

# Appendix 1

**Appendix 1—table 1.** Overview of the musical stimuli presented in the MEG (top) and EEG study (bottom).

**MusicMEG**

| Composer | Composition | Year | Key | Time signature | Tempo (bpm) | Duration (sec) | Notes | Sound |
|---|---|---|---|---|---|---|---|---|
| Benjamin Britten | Metamorphoses Op. 49, II. Phaeton | 1951 | C maj | 4/4 | 110 | 95 | 384 | Oboe |
| Benjamin Britten | Metamorphoses Op. 49, III. Niobe | 1951 | Db maj | 4/4 | 60 | 101 | 171 | Oboe |
| Benjamin Britten | Metamorphoses Op. 49, IV. Bacchus | 1951 | F maj | 4/4 | 100 | 114 | 448 | Oboe |
| César Franck | Violin Sonata IV. Allegretto poco mosso | 1886 | A maj | 4/4 | 150 | 175 | 458 | Flute |
| Carl Philipp Emanuel Bach | Sonata for Solo Flute, Wq.132/H.564 III. | 1763 | A min | 3/8 | 98 | 275 | 1358 | Flute |
| Ernesto Köhler | Flute Exercises Op. 33 a, V. Allegretto | 1880 | G maj | 4/4 | 124 | 140 | 443 | Flute |
| Ernesto Köhler | Flute Exercises Op. 33b, VI. Presto | 1880 | D min | 6/8 | 176 | 134 | 664 | Piano |
| Georg Friedrich Händel | Flute Sonata Op. 1 No. 5, HWV 363b, IV. Bourrée | 1711 | G maj | 4/4 | 132 | 84 | 244 | Oboe |
| Georg Friedrich Händel | Flute Sonata Op. 1 No. 3, HWV 379, IV. Allegro | 1711 | E min | 3/8 | 96 | 143 | 736 | Piano |
| Joseph Haydn | Little Serenade | 1785 | F maj | 3/4 | 92 | 81 | 160 | Oboe |
| Johann Sebastian Bach | Flute Partita BWV 1013, II. Courante | 1723 | A min | 3/4 | 64 | 176 | 669 | Flute |
| Johann Sebastian Bach | Flute Partita BWV 1013, IV. Bourrée angloise | 1723 | A min | 2/4 | 62 | 138 | 412 | Oboe |
| Johann Sebastian Bach | Violin Concerto BWV 1042, I. Allegro | 1718 | E maj | 2/2 | 100 | 122 | 698 | Piano |
| Johann Sebastian Bach | Violin Concerto BWV 1042, III. Allegro Assai | 1718 | E maj | 3/8 | 92 | 80 | 413 | Piano |
| Ludwig van Beethoven | Sonatina (Anh. 5 No. 1) | 1807 | G maj | 4/4 | 128 | 210 | 624 | Flute |
| Muzio Clementi | Sonatina Op. 36 No. 5, III. Rondo | 1797 | G maj | 2/4 | 112 | 187 | 915 | Piano |
| Modest Mussorgsky | Pictures at an Exhibition - Promenade | 1874 | Bb maj | 5/4 | 80 | 106 | 179 | Oboe |
| Pyotr Ilyich Tchaikovsky | The Nutcracker Suite - Russian Dance Trepak | 1892 | G maj | 2/4 | 120 | 78 | 396 | Piano |
| Wolfgang Amadeus Mozart | The Magic Flute K620, Papageno's Aria | 1791 | F maj | 2/4 | 72 | 150 | 452 | Flute |
| | | | | | | 2589 | 9824 | |

**MusicEEG**

| Composer | Composition | Year | Key | Time signature | Tempo (bpm) | Duration (sec) | Notes | Sound |
|---|---|---|---|---|---|---|---|---|
| Johann Sebastian Bach | Flute Partita BWV 1013, I. Allemande | 1723 | A min | 4/4 | 100 | 158 | 1022 | Piano |
| Johann Sebastian Bach | Flute Partita BWV 1013, II. Corrente | 1723 | A min | 3/4 | 100 | 154 | 891 | Piano |
| Johann Sebastian Bach | Flute Partita BWV 1013, III. Sarabande | 1723 | A min | 3/4 | 70 | 120 | 301 | Piano |
| Johann Sebastian Bach | Flute Partita BWV 1013, IV. Bourree | 1723 | A min | 2/4 | 80 | 135 | 529 | Piano |
| Johann Sebastian Bach | Violin Partita BWV 1004, I. Allemande | 1723 | D min | 4/4 | 47 | 165 | 540 | Piano |
| Johann Sebastian Bach | Violin Sonata BWV 1001, IV. Presto | 1720 | G min | 3/8 | 125 | 199 | 1604 | Piano |
| Johann Sebastian Bach | Violin Partita BWV 1002, I. Allemande | 1720 | Bb min | 4/4 | 50 | 173 | 620 | Piano |
| Johann Sebastian Bach | Violin Partita BWV 1004, IV. Gigue | 1723 | D min | 12/8_ | 120 | 182 | 1352 | Piano |
| Johann Sebastian Bach | Violin Partita BWV 1006, II. Loure | 1720 | E maj | 6/4 | 80 | 134 | 338 | Piano |
| Johann Sebastian Bach | Violin Partita BWV 1006, III. Gavotte | 1720 | E maj | 4/4 | 140 | 178 | 642 | Piano |
| | | | | | | 1598 | 7839 | |

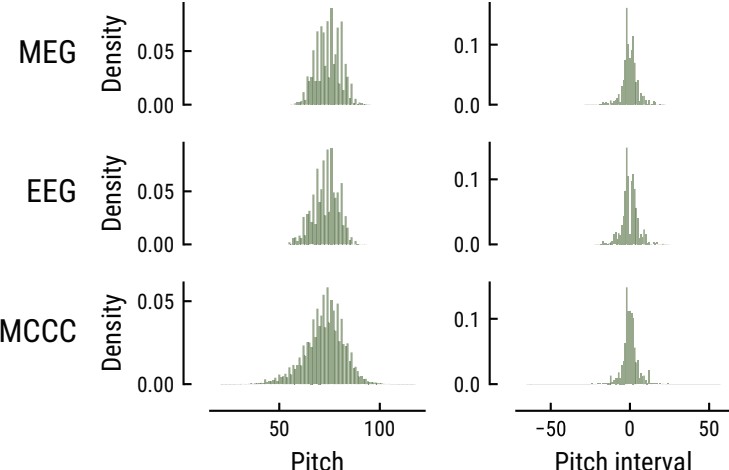

**Appendix 1—figure 1.** Comparison of the pitch (left) and pitch interval distributions (right) for the music data from the MEG study (top), EEG study (middle), and MCCC corpus (bottom).

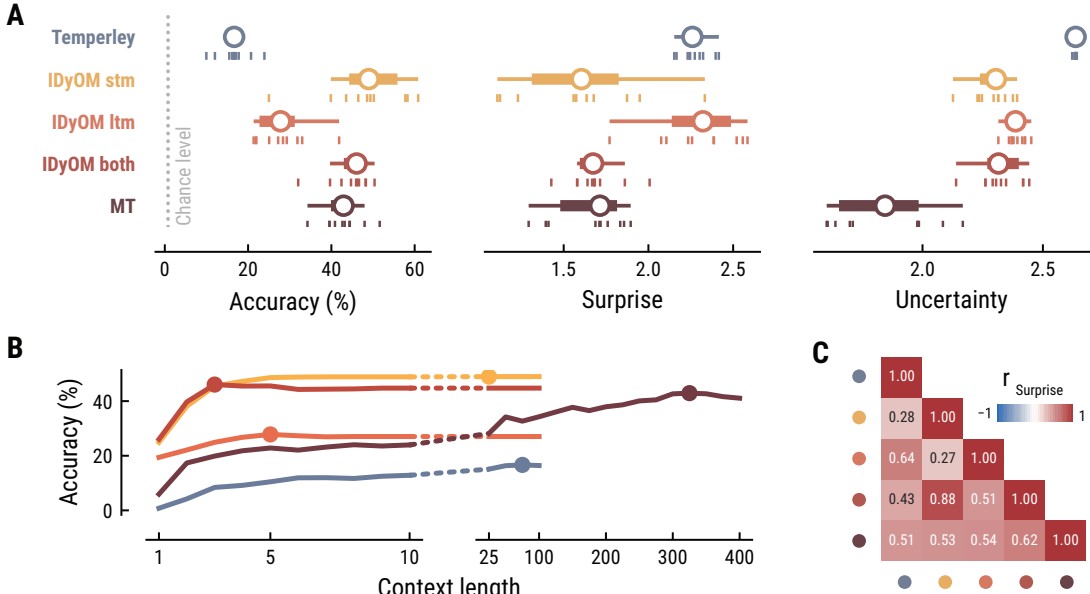

**Appendix 1—figure 2.** Model performance on the musical stimuli used in the EEG study. (**A**) Comparison of music model performance in predicting upcoming note pitch, as composition-level accuracy (left; higher is better), median surprise across notes (middle; lower is better), and median uncertainty across notes (right). Context length for each model is the best performing one across the range shown in (**B**). Vertical bars: single compositions, circle: median, thick line: quartiles, thin line: quartiles ±1.5 × interquartile range. (**B**) Accuracy of note pitch predictions (median across 10 compositions) as a function of context length and model class (same color code as (**A**)). Dots represent maximum for each model class. (**C**) Correlations between the surprise estimates from the best models.

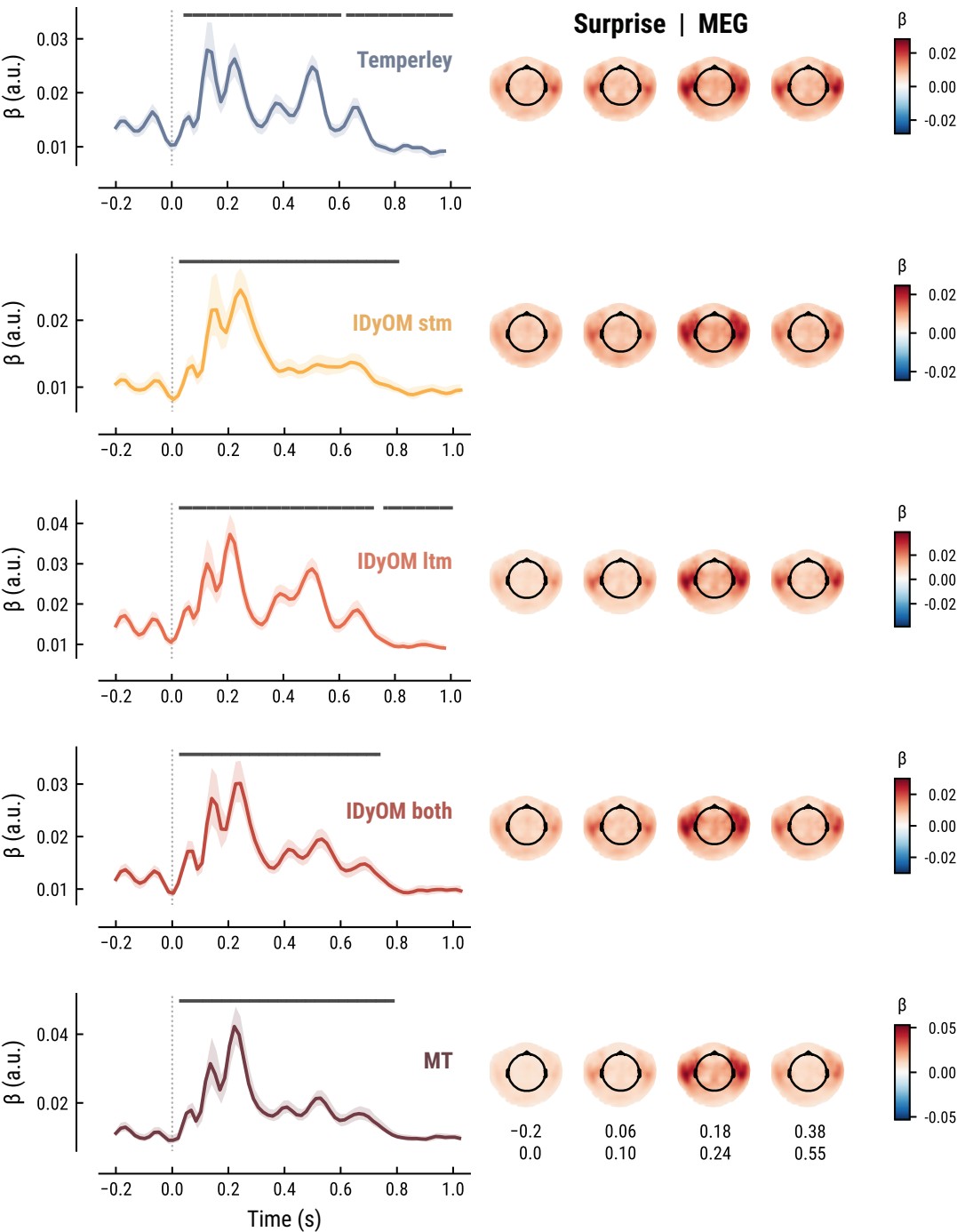

**Appendix 1—figure 3.** Comparison of the MEG TRFs and spatial topographies for the surprise estimates from the best models of each model class.

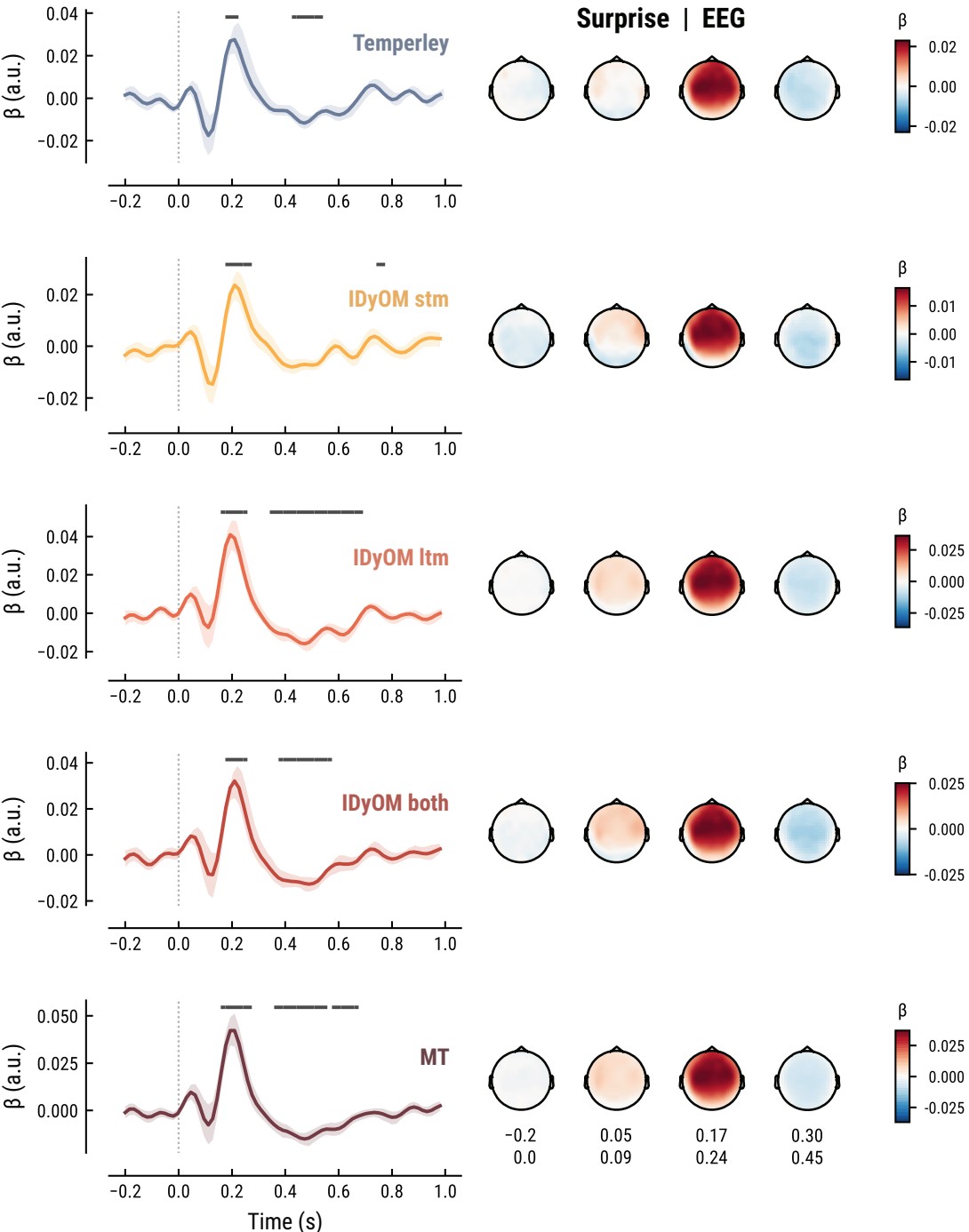

**Appendix 1—figure 4.** Comparison of the EEG TRFs and spatial topographies for the surprise estimates from the best models of each model class.

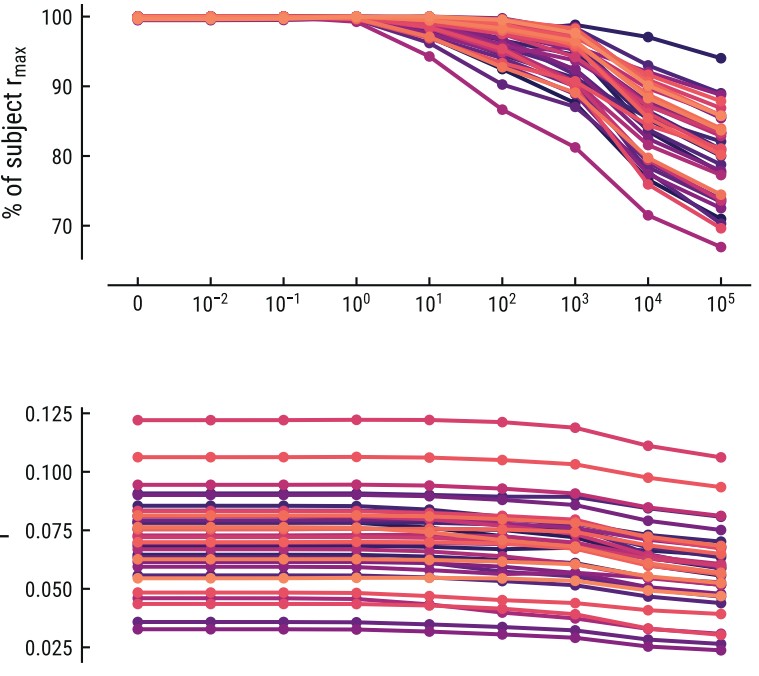

**Appendix 1—figure 5.** Comparison of the predictive performance on the MEG data using ridge-regularized regression, with the optimal cost hyperparameter alpha estimated using nested cross-validation. Results are shown for the best-performing model (MT, context length of 8 notes). Each line represents one participant. Lower panel: raw predictive performance (**r**). Upper panel: predictive performance expressed as percentage of a participant's maximum.

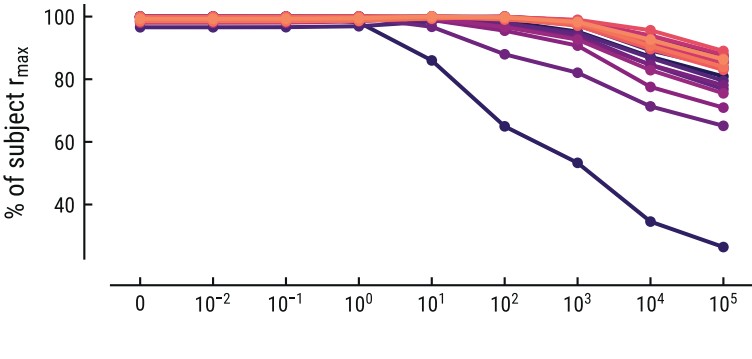

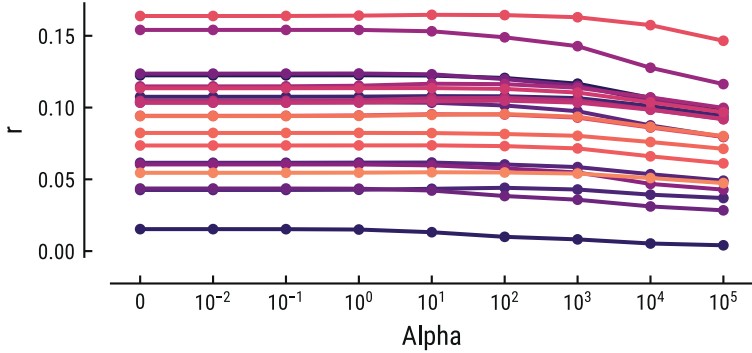

**Appendix 1—figure 6.** Comparison of the predictive performance on the EEG data using ridge-regularized regression, with the optimal cost hyperparameter alpha estimated using nested cross-validation. Results are shown for the best-performing model (MT, context length of 7 notes). Each line represents one participant. Lower panel: raw predictive performance (**r**). Upper panel: predictive performance expressed as percentage of a participant's maximum.

