## [Editor Report]

This study models the predictions a listener makes in music in two ways: how different model algorithms compare in their performance at predicting the upcoming notes in a melody, and how well they predict listeners' brain responses to these notes. The study will be important as it implements and compares three contemporary models of music prediction. In a set of convincing analyses, the authors find that musical melodies are best predicted by models taking into account long-term experience of musical melodies, whereas brain responses are best predicted by applying these models to only a few most recent notes.

---

## [Decision Letter]

**Decision letter after peer review:**

Thank you for submitting your article "Cortical activity during naturalistic music listening reflects short-range predictions based on long-term experience" for consideration by *eLife*. Your article has been reviewed by 3 peer reviewers, and the evaluation has been overseen by a Reviewing Editor and Christian Büchel as the Senior Editor. The following individuals involved in the review of your submission have agreed to reveal their identity: William Sedley (Reviewer #2); Keith Doelling (Reviewer #3).

Essential revisions:

1) The one non-standard feature of the analysis is the lack of regularization (e.g., ridge). The authors should perform the analysis using regularization (via nested cross-validation) and test if the predictions are improved (if they have already done this, they should report the results).

2) The authors make a distinction between "Gestalt-like principles" and "statistical learning" but they never define was is meant by this distinction. The Temperley model encodes a variety of important statistics of Western music, including statistics such as keys that are unlikely to reflect generic Gestalt principles. The Temperley model builds in some additional structure such as the notion of a key, which the n-gram and transformer models must learn from scratch. In general, the models being compared differ in so many ways that it is hard to conclude much about what is driving the observed differences in prediction accuracy, particularly given the small effect sizes. The context manipulation is more controlled, and the fact that neural prediction accuracy dissociates from the model performance is potentially interesting. However, we were not confident that the authors have a good neural index of surprise for the reasons described above, and this might limit the conclusions that can be drawn from this manipulation in the manuscript as is.

3) The authors may overstate the advancement of the Music Transformer with the present stimuli, as its increase in performance requires a considerably longer context than the other models. Secondly, the Baseline model, to which the other models are compared, does not contain any pitch information on which these models operate on. As such, it's unclear if the advancements of these models come from being based on new information or the operations it performs on this information as claimed.

4) Source analysis: See below in Rev #1 and Rev #3 for concerns over the results and interpretation of the source localisation.

*Reviewer #1 (Recommendations for the authors):*

The one non-standard feature of the analysis is the lack of regularization (e.g., ridge). The authors should perform the analysis using regularization (via nested cross-validation) and test if the predictions are improved (if they have already done this, they should report the results).

Source localization analysis with MEG or EEG is highly uncertain. The conclusion that the results are coming from "fronto-temporal" areas doesn't strike me as adding much value; I'd be inclined to remove this analysis from the manuscript. Minimally, the authors should note that source localization is highly uncertain.

The authors should report the earphones/tubes used to present sounds.

*Reviewer #3 (Recommendations for the authors):*

– Figure 2: Music Transformer is presented as the state-of-the-art model throughout the paper, with the main advantage of grasping regularities on longer time scales. Yet the computational results in figure 2 tend to show that it does not bring much in terms of musical predictions compared to IDyOM. MT also needs a way larger context length to reach the same accuracy. It has a lower uncertainty, but this feature did not improve the results of the neural analysis. This point could be discussed to better understand why MT is a better model of human musical expectations and surprise.

– The source analysis is a bit puzzling to me. The results show that every feature (including Note Onset and Note Repetition) are localized in Broca's area, frontally located. Wouldn't the authors expect that a feature as fundamental as Note Onset should be clearly (or at least partially) localized in the primary auditory cortex? Because these results localize frontally, it is hard to fully trust the MT surprise results as well. Can the authors provide some form of sanity check to provide further clarity on these results? Perhaps the source localizing the M100 to show the analysis pipeline is not biased towards frontal areas? Do the authors expect note onsets in continuous music to be represented in higher-order areas than the representation of a single tone? Figure 7 and source-level results: the authors do not discuss the fact that the results are mainly found in a frontal area. It looks like there is no effect in the auditory cortex, which is surprising and should be discussed.

– The dipole fit shows a high correlation with behavior but it is not compared with any alternatives. Can the authors use some model comparison methods (e.g. AIC or BIC) to show that a single dipole per hemisphere is a better fit than two or three?

– Line 252: the order in which the transformation from axial to planar gradients is applied with respect to other processing steps (e.g., z-scoring of the MEG data and TRF fitting) is not clear. Is it the MEG data that was transformed as suggested here, or is the transformation applied to TRFs coefficients (as explained in line 710), with TRFs trained on axial gradiometers? This has downstream consequences that lead to a lack of clarity in the results. For example, the MEG data shows positive values in the repetition kernel which if the transformation was applied before the TRF would imply that repetition increases activity rather than reduces it as would be expected. From this, I infer that it is the TRF kernels that were transformed. I recognize that the authors can infer results regarding directionality in the EEG data but this analysis choice results in an unnecessary loss of information in the MEG analysis. Please clarify the methods and if the gradiometer transformation is performed on the kernels, I would recommend re-running the analysis with gradiometer transformation first.

– The Baseline model claims to include all relevant acoustic information but it does not include the note pitches themselves. As these pitches are provided to the model, it is difficult to tell whether the effects of surprise are related to model output predictions, or how well they represent their input. If the benefits from the models rely truly on their predictive aspect then including pitch information in the baseline model would be an important control. The authors have done their analysis in a way that fits what previous work has done but I think it is a good time to correct this error.

– I wonder if the authors could discuss a bit more about the self-attention component of the Music Transformer model. I'm not familiar with the model's inner workings but I am intrigued by this dichotomy of a larger context to improve musical predictions and a shorter context to improve neural predictions. I wonder if a part of this dissociation has to do with the self-attention feature of Transformers, that a larger context is needed to have a range of motifs to draw from but the size of the motifs that the model attends to should be of a certain size to fit neural predictions.

– Figure 3C: it could be interesting to show the same figure for IDyOM ltm. Given that context length does not impact ltm as much as MT, we could obtain different results. Since IDyOM ltm gets similar results to MT on the MEG data (figure 3A, no significant difference), it is thus hard to tell if the influence of context length comes from brain processing or the way MT works.

– Figure 8: Adding the uncertainty estimates did not improve the model's predictive performance compared to surprise alone, but what about TRFs trained on uncertainty without surprise? Without this result, it is hard to understand why the surprise was chosen over uncertainty.

– The sentence from lines 272 to 274 is not clear. In particular, the late negativity effect seems to be present in EEG data only, and it is thus hard to understand why a negative correlation between surprise estimates of subsequent notes would have such an effect in EEG and not MEG. Moreover, the same late negativity effect can be seen on the TRF of note onset but is not discussed.

– Some of the choices for the Temperley Model seem to unnecessarily oversimplify and restrict its potential performance. In the second principle, the model only assesses the relationships between neighboring notes when the preceding note is the tonal centroid. It would seem more prudent to include all notes to collect further data. In the third principle, the model marginalizes major and minor scales by weighting probabilities of each profile by the frequency of major and minor pieces in the database. Presumably, listeners can identify the minor or major key of the current piece (at least implicitly). Why not select the model for each piece, outright?

– Stimulus: Many small details make it so that the stimuli are not so naturalistic (MIDI velocity set to 100, monophonic, mono channel…). This results in a more controlled experiment, but the claim that it expands melodic expectation findings to naturalistic music listening is a bit bold.

– Line 478: authors refer to "original compositions" which may give the impression that the pieces were written for the experiment. From the rest of the text, I don't believe this to be true.

– Formula line 553: the first probability of Xt (on the left) should also be a conditional probability of Xt given previous values of x. This is the entropy of the probability distribution estimated by the model.

– Line 667: the study by Di Liberto from which the EEG data come uses a ridge regression (ridge regularization). Is there a reason to use a non-regularized regression in this case? This should be discussed in the methods.

---

## [Author Response]

Essential revisions:1) The one non-standard feature of the analysis is the lack of regularization (e.g., ridge). The authors should perform the analysis using regularization (via nested cross-validation) and test if the predictions are improved (if they have already done this, they should report the results).

Our motivation for the ‘plain’ ordinary least squares (OLS) was, firstly, that we use the regression ERP/F modelling framework (Smith and Kutas, 2015). This means that our design matrices were very sparse, with little correlation between the time-shifted regressors and hence a comparatively low risk for overfitting. Moreover, the model comparisons were always performed in a cross-validated fashion (thus any potential overfitting would reduce, rather than artificially inflate, model performance). However, we appreciate that we hereby deviate from earlier similar work, which used ridgeregularized regression.

We therefore now also implemented ridge-regularized regression, with the optimal cost hyperparameter α estimated using nested cross-validation. The results for this are shown in Appendix—Figure 5 (one curve per participant; regression of best-performing model (MT, context length 8) on MEG data):

This clearly demonstrates that the OLS we used previously (α = 0) is always among the best-fitting models. Even for those participants that show an increase in cross-validated r for non-zero regularization, these increases are negligible. We therefore report the same OLS models in the manuscript as before, and have now added the above figure to the supplemental information.

This also holds for the EEG data. Appendix—figure 6 shows the results for the best-performing model (MT, context length 7).

2) The authors make a distinction between "Gestalt-like principles" and "statistical learning" but they never define was is meant by this distinction. The Temperley model encodes a variety of important statistics of Western music, including statistics such as keys that are unlikely to reflect generic Gestalt principles. The Temperley model builds in some additional structure such as the notion of a key, which the n-gram and transformer models must learn from scratch. In general, the models being compared differ in so many ways that it is hard to conclude much about what is driving the observed differences in prediction accuracy, particularly given the small effect sizes. The context manipulation is more controlled, and the fact that neural prediction accuracy dissociates from the model performance is potentially interesting. However, we were not confident that the authors have a good neural index of surprise for the reasons described above, and this might limit the conclusions that can be drawn from this manipulation in the manuscript as is.

First of all, we would like to apologize for any unclarity regarding the distinction between Gestalt-like and statistical models. We take Gestalt-like models to be those that explain music perception as following a restricted set of rules, such as that adjacent notes tend to be close in pitch. In contrast, as the reviewer correctly points out, statistical learning models have no such *a priori* principles and must learn similar or other principles from scratch. Importantly, the distinction between these two classes of models is not one we make for the first time in the context of music perception. Gestalt-like models have a long tradition in musicology and the study of music cognition dating back to (Meyer, 1957). The Implication-Realization model developed by Eugene Narmour (Narmour, 1990, 1992; Schellenberg, 1997) is another example for a rule-based theory of music listening, which has influenced the model by David Temperley, which we applied as the most recently influential Gestalt-model of melodic expectations in the present study. Concurrently to the development of Gestalt-like models, a second strand of research framed music listening in light of information theory and statistical learning (Bharucha, 1987; Cohen, 1962; Conklin and Witten, 1995; Pearce and Wiggins, 2012). Previous work has made the same distinction and compared models of music along the same axis (Krumhansl, 2015; Morgan et al., 2019a; Temperley, 2014). We have updated the manuscript to elaborate on this distinction and highlight that it is not uncommon.

Second, we emphasize that we compare the models directly in terms of their predictive performance both of upcoming musical notes and of neural responses. This predictive performance is not dependent on the internal details of any particular model; e.g. in principle it would be possible to include a “human expert” model where we ask professional composers to predict upcoming notes given a previous context. Because of this independence of the relevant comparison metric on model details, we believe comparing the models is justified. Again, this is in line with previously published work in music (Morgan et al., 2019a), language, (Heilbron et al., 2022; Schmitt et al., 2021; Wilcox et al., 2020), and other domains (Planton et al., 2021). Such work compares different models in how well they align with human statistical expectations by assessing how well different models explain predictability/surprise effects in behavioral and/or brain responses.

Third, regarding the doubts on the neural index of surprise used: we respond to this concern below, after reviewer 1’s first point to which the present comment refers (the referred-to comment was not included in the “Essential revisions” here).

3) The authors may overstate the advancement of the Music Transformer with the present stimuli, as its increase in performance requires a considerably longer context than the other models.

We do not believe to have overstated the advance presented by the MusicTransformer, for the following reasons. First, we appreciate that from the music analysis (Figure 2b and Figure A3b), it seems as if the Music Transformer requires much longer context to reach only slightly higher predictive performance on the musical stimuli. Note, however, that this only applies to the comparison between MT and the IDyOM-stm and IDyOM-both (which subsumes IDyOM-stm) models, but not to the comparison between MT and IDyOM-ltm or the Temperley model. The MT and IDyOM-stm (and therefore IDyOM-both) deal with context information rather differently, possibly leading to the wrong impression that predictive performance for the MT requires a lot more ‘data’. We go into these differences in more detail below.

Second, and importantly, the distinctive empirical contribution of our study is not the superiority of the MT over the other models per se, but the (neural and predictive) performance differences among model classes: statistical learning (IDyOM/MT) versus Gestalt/rule-based (Temperley), and the dependence of performance on context lengths. For these comparisons, the MT is a very useful tool because it efficiently tracks hierarchical structure in longer musical contexts (Huang et al., 2018). We furthermore demonstrate that it works at least as well as the previous state-of-the-art statistical model (IDyOM), yet may process a much larger class of music (i.e., polyphonic music; not yet explored).

Regarding the first point: There are small technical differences in the way previous context is used by IDyOM-stm and the MusicTransformer (MT). IDyOM-stm is an n-gram model, predicting the probability of an upcoming note x_t_ given a previous context {x_t-1_…x_t-n_}. The context parameter we varied here governs the maximum length n of n-grams that IDyOM can take into account to make its predictions. Importantly, IDyOM-stm is an on-line statistical learning model: it updates the relative probabilities p(x_t_ | {x_t-1_…x_t-n_}) as it is making predictions and parsing the ongoing composition. So while for any given note IDyOM-stm will only directly take into account the preceding n notes, the underlying statistical model against which it will interpret those n context notes can depend on all the n-grams and subsequent notes that preceded it. Because of this property, IDyOM-stm is in effect “learning” based on the current ongoing composition and therefore can indirectly leverage more information than the strict limit of n-grams considered. (It could be said that IDyOM-stm is ‘peeking’ at the test set to some extent, and therefore its predictive performance may be slightly overestimated.) Importantly, the type of on-line updating in IDyOM-stm still precludes the learning of any hierarchical structure encompassing longer context lengths than n (which is, for our purposes, an essential difference with the MT).

The MT model performs no on-line learning. Instead, the model only takes into account the strict n context notes that it is provided with when asked for an upcoming prediction. Critically, the transformer architecture enables the MT to make hierarchical predictions on those n notes, which depend on the musical corpus it was trained on.

Finally, regarding the second point: We further note that the dependence of predictive performance on context length is quite different between predicting music (Figure 2) and predicting neural responses (Figures 3 and 4). For predicting upcoming musical notes, indeed the MT required considerably larger context lengths in order to outperform IDyOM (stm and both), likely in part due to the reasons described above. In contrast, for the related surprise scores to predict neural responses, the context length required for the MT to peak was on the same order as IDyOM.

Secondly, the Baseline model, to which the other models are compared, does not contain any pitch information on which these models operate on. As such, it's unclear if the advancements of these models come from being based on new information or the operations it performs on this information as claimed.

We apologize for not being clear enough here. Importantly, none of the models compared contained any exact pitch information. We only used surprisal (and uncertainty) collapsed across the entire distribution of possible upcoming notes as a regressor for the MEG and EEG data, which by itself cannot be traced back to any particular pitch. Furthermore, we already did include a confound regressor encoding low versus high pitch, which, critically, was included identically in all the models (including the Baseline). We have updated the manuscript to emphasize this point.

4) Source analysis: See below in Rev #1 and Rev #3 for concerns over the results and interpretation of the source localisation.

In response to the reviewers’ comments, we revisited the source analysis considerably. Previously, we had only investigated the later peaks, for which we had no very strong expectations and hence the frontal peak did not appear particularly suspect. However, as a sanity check and as suggested by reviewer 3, we source localized the earliest peak of the Onset TRF, expecting a clear bilateral peak in early auditory cortex. In contrast, this peak was also localized to frontal cortex, which raised our suspicions that something was wrong in our source analysis pipeline. We indeed identified a bug: for all participants, the pipeline used the forward model computed for one participant, rather than each individual participant’s forward model. If this one participant happened to sit somewhat further to the front or back of the MEG helmet than the mean, and/or have a different head size than the mean, this would introduce a consistent spatial bias, which could explain the previous absence of a peak in auditory cortex. We apologize for this error in our code, and are intensely grateful to the reviewers for their well-founded suspicion.

We have now re-run the source analysis using the correct forward models, and additionally compare three models with one to three dipoles per hemisphere, as suggested by reviewer 3. As the final solution per participant, we use that with the largest adjusted-r^2^ score in explaining the observed sensor topography. For the majority of participants, the three-dipole model fits best, although the gain in adjusted-r^2^ over the one- and two-dipole models is modest:

**Author response image 1. sa2fig1:** 

(Even for the one-dipole model, these adjusted-r^2^ scores are higher than the previously reported r^2^, since now the correct forward models are used.) Using this corrected and updated pipeline, we now find consistent peaks in auditory cortex for all three regressors, also in the later time window of interest, and have updated the manuscript accordingly.Additionally, in accordance with reviewer 1’s suggestion, we have now additionally reflected on the limits of MEG’s spatial resolution in the Discussion.

Reviewer #1 (Recommendations for the authors):The one non-standard feature of the analysis is the lack of regularization (e.g., ridge). The authors should perform the analysis using regularization (via nested cross-validation) and test if the predictions are improved (if they have already done this, they should report the results).

See our response in “Essential revisions” above.

Source localization analysis with MEG or EEG is highly uncertain. The conclusion that the results are coming from "fronto-temporal" areas doesn't strike me as adding much value; I'd be inclined to remove this analysis from the manuscript. Minimally, the authors should note that source localization is highly uncertain.

See our response in “Essential revisions” above.

The authors should report the earphones/tubes used to present sounds.

The earmolds used to present sounds were Doc’s Promolds NonVent with #13 thick prebent 1.85 mm ID tubes, from Audine Healthcare, in combination with Etymotic ER3A earphones. We have now added this information to the manuscript.

Reviewer #3 (Recommendations for the authors):– Figure 2: Music Transformer is presented as the state-of-the-art model throughout the paper, with the main advantage of grasping regularities on longer time scales. Yet the computational results in figure 2 tend to show that it does not bring much in terms of musical predictions compared to IDyOM. MT also needs a way larger context length to reach the same accuracy. It has a lower uncertainty, but this feature did not improve the results of the neural analysis. This point could be discussed to better understand why MT is a better model of human musical expectations and surprise.

See our response in “Essential revisions” above.

– The source analysis is a bit puzzling to me. The results show that every feature (including Note Onset and Note Repetition) are localized in Broca's area, frontally located. Wouldn't the authors expect that a feature as fundamental as Note Onset should be clearly (or at least partially) localized in the primary auditory cortex? Because these results localize frontally, it is hard to fully trust the MT surprise results as well. Can the authors provide some form of sanity check to provide further clarity on these results? Perhaps the source localizing the M100 to show the analysis pipeline is not biased towards frontal areas? Do the authors expect note onsets in continuous music to be represented in higher-order areas than the representation of a single tone? Figure 7 and source-level results: the authors do not discuss the fact that the results are mainly found in a frontal area. It looks like there is no effect in the auditory cortex, which is surprising and should be discussed.

See our response in “Essential revisions” above.

– The dipole fit shows a high correlation with behavior but it is not compared with any alternatives. Can the authors use some model comparison methods (e.g. AIC or BIC) to show that a single dipole per hemisphere is a better fit than two or three?

See our response in “Essential revisions” above.

– Line 252: the order in which the transformation from axial to planar gradients is applied with respect to other processing steps (e.g., z-scoring of the MEG data and TRF fitting) is not clear. Is it the MEG data that was transformed as suggested here, or is the transformation applied to TRFs coefficients (as explained in line 710), with TRFs trained on axial gradiometers? This has downstream consequences that lead to a lack of clarity in the results. For example, the MEG data shows positive values in the repetition kernel which if the transformation was applied before the TRF would imply that repetition increases activity rather than reduces it as would be expected. From this, I infer that it is the TRF kernels that were transformed. I recognize that the authors can infer results regarding directionality in the EEG data but this analysis choice results in an unnecessary loss of information in the MEG analysis. Please clarify the methods and if the gradiometer transformation is performed on the kernels, I would recommend re-running the analysis with gradiometer transformation first.

Indeed, we estimated the TRFs on the original axial gradient data and subsequently (1) (linearly) transformed those axial TRFs to planar gradient data and (2) (nonlinearly) combined the resulting synthetic horizontal and vertical planar gradient components to a single magnitude per original MEG sensor. This has, first of all, the advantage that we can perform source analysis straightforwardly on the axial-gradient TRFs (analogous to an axial-gradient ERF). Most importantly, however, this order of operations prevents the amplification of noise that would result from executing the non-linear combination step (2) on the continuous, raw MEG data. For this latter reason, this order of operations is the de facto standard in ERF research (as well as in other recent studies employing TRFs or “regression ERFs”).

Estimating the TRFs on the non-combined synthetic planar gradient data (so after step 1, but before step 2) would not suffer from this noise amplification issue. However, since step 1 is linear, this would yield exactly the same results as the current order of operations, while doubling the computational cost of the regression.

We apologize for being unclear on the exact order of operations regarding the planar gradient transformation in the original manuscript and have now clarified this.

– The Baseline model claims to include all relevant acoustic information but it does not include the note pitches themselves. As these pitches are provided to the model, it is difficult to tell whether the effects of surprise are related to model output predictions, or how well they represent their input. If the benefits from the models rely truly on their predictive aspect then including pitch information in the baseline model would be an important control. The authors have done their analysis in a way that fits what previous work has done but I think it is a good time to correct this error.

See our response in “Essential revisions” above.

– I wonder if the authors could discuss a bit more about the self-attention component of the Music Transformer model. I'm not familiar with the model's inner workings but I am intrigued by this dichotomy of a larger context to improve musical predictions and a shorter context to improve neural predictions. I wonder if a part of this dissociation has to do with the self-attention feature of Transformers, that a larger context is needed to have a range of motifs to draw from but the size of the motifs that the model attends to should be of a certain size to fit neural predictions.

We were similarly intrigued by this dissociation; however, we believe the most likely explanation of the dissociation is neural in origin, rather than reflecting the self-attention mechanism. This mechanism indeed allows the transformer to “attend” preceding notes in various ways: a prediction might, for example, be based on a sequence of several preceding notes, while taking into account several notes (or motifs) from many notes ago, but nothing in between. We thus agree that large contexts allow the MT to “have a range of motifs to draw from”. This is a good way to put it, since the model itself ‘decides’ which part of the context to ‘draw from’.

However, we do not think this can explain the dissociation. Instead, we observe that increasing the context ‘seen’ by the MT steadily improves its predictions (Figure 2b), and we suggest that at some point (over 5-10 notes of context), the MT expectations become more sophisticated than the expectations reflected in the MEG signal. This does not mean they are more sophisticated than human listeners’ expectations: humans can clearly track and appreciate patterns in music much longer and richer than the MT can. However, it seems that such high-level, long-range, likely hierarchical expectations are not driving the surprise effects in the evoked responses, which instead seem to reflect more low-level predictions over shorter temporal scales. The neural processing of the highest-level, longest-range predictions are likely not time-locked to the onset of musical notes, which precludes these being detected with the techniques used in the present study.

Finally, another reason to believe that the dissociation between context-versus-music-prediction and context-versus-brain-response is not driven by the specific details of the MT is that the same dissociation is observed for the IDyOM models. There, the pattern is much less clear because the musical prediction performance plateaus early, such that the musical predictions never become ‘much smarter’ than the neural predictions. The same pattern is nonetheless observed: musical prediction continues to improve for longer contexts than the neural prediction.

– Figure 3C: it could be interesting to show the same figure for IDyOM ltm. Given that context length does not impact ltm as much as MT, we could obtain different results. Since IDyOM ltm gets similar results to MT on the MEG data (figure 3A, no significant difference), it is thus hard to tell if the influence of context length comes from brain processing or the way MT works.

In Author response image 2 we have added a plot showing the predictive performance of the IDyOM ltm model on MEG versus music data similar to Figure 3C for the Music Transformer. We believe the MT to be the more sensitive measure of context dependency, for reasons outlined earlier. For that reason, we have decided not to add it to the manuscript. Furthermore, even though this particular plot is not included, the exact same traces feature as part of Figures 2B and 3B, so the information is present in the manuscript nonetheless.

– Figure 8: Adding the uncertainty estimates did not improve the model's predictive performance compared to surprise alone, but what about TRFs trained on uncertainty without surprise? Without this result, it is hard to understand why the surprise was chosen over uncertainty.

We did not investigate regression models using only the uncertainty regressor for two reasons. First, we were *a priori* primarily interested in the neural response to surprise, rather than uncertainty. Surprise is a much more direct index of content-based expectations and their violation than (unspecific) uncertainty, and since our theoretical interest is in content-based expectations, we focused on the former.

Second, we did explore the effect of uncertainty as a secondary interest, but found that adding uncertainty to the regression model not only did not improve the cross-validated performance, but actually *worsened* it (Figure 8B). Surprise and uncertainty were modestly correlated (Figure 8A), and therefore the most likely interpretation of this drop in cross-validated performance is that uncertainty truly does not explain additional neural variance. (That is, any neural variance it would explain on its own is likely due to its correlation with surprise; if it were to capture unique neural variance by itself, then the performance of the joint model would be at least as high as the model featuring only surprise.) For this *a posteriori* reason, in addition to the *a priori* reason already formulated, we did not further explore regression models featuring only uncertainty.

– The sentence from lines 272 to 274 is not clear. In particular, the late negativity effect seems to be present in EEG data only, and it is thus hard to understand why a negative correlation between surprise estimates of subsequent notes would have such an effect in EEG and not MEG. Moreover, the same late negativity effect can be seen on the TRF of note onset but is not discussed.

We apologize for the unclarity here. We emphasize that any judgements regarding the polarity of the effects are based on the EEG data (Figure 6), as well as inspection of the axial-gradient MEG data (not shown). This was already made explicit in the manuscript around lines 259-261. We have now rephrased the relevant passage. We hope that this should alleviate any worry of a discrepancy between the MEG and EEG results.

Regarding the same late negativity for the Onset regressor: we do already emphasize the presence of this deflection (line 268), but since our interest is in the modulation of neural response by surprise (and, to a lesser extent, repetitions, which are related to surprise), we do not reflect on it in any further detail. Note that the presence of this deflection in the Onset TRF does not lessen the importance of its presence in the Surprise TRF – the latter remains indication of this particular peak being modulated by musical surprise.

– Some of the choices for the Temperley Model seem to unnecessarily oversimplify and restrict its potential performance. In the second principle, the model only assesses the relationships between neighboring notes when the preceding note is the tonal centroid. It would seem more prudent to include all notes to collect further data. In the third principle, the model marginalizes major and minor scales by weighting probabilities of each profile by the frequency of major and minor pieces in the database. Presumably, listeners can identify the minor or major key of the current piece (at least implicitly). Why not select the model for each piece, outright?

We used the Temperley model as David Temperley and others have formulated and applied it in previous research. While some of these choices could indeed be debated, we aimed to use the model in line with the existing literature, which has demonstrated the capabilities of the model in a variety of tasks, such as pitch prediction, key finding, or explaining behavioural ratings of melodic surprise (Morgan et al., 2019b; Temperley, 2008, 2014). We have now explicitly mentioned that the specifics in the three principles were chosen in accordance with earlier work.

– Stimulus: Many small details make it so that the stimuli are not so naturalistic (MIDI velocity set to 100, monophonic, mono channel…). This results in a more controlled experiment, but the claim that it expands melodic expectation findings to naturalistic music listening is a bit bold.

We agree, and do not wish to make the claim that the stimuli we used are representative of the full breadth of music that humans may encounter in everyday life. However, we do maintain that these stimuli are considerably closer to naturalistic music than much previous work on the neural basis of (the role of expectations in) music processing. It could be argued that the most severe limitation to a broad claim of ‘naturalistic’ is the use of strictly monophonic music. This was a technical necessity given two of the three model classes (IDyOM, Temperley). An important contribution of our work is to demonstrate that a different model (MusicTransformer) does at least equally well as the previous state-of-the-art. Critically, the MT support the processing of polyphonic music, and our work thus paves the way for future studies investigating neural expectations in music more representative of that which is encountered in daily life. In accordance with the reviewer’s suggestion, we have now nuanced our claim of ‘naturalistic’ in the Discussion.

– Line 478: authors refer to "original compositions" which may give the impression that the pieces were written for the experiment. From the rest of the text, I don't believe this to be true.

The reviewer is correct; this is now fixed.

– Formula line 553: the first probability of Xt (on the left) should also be a conditional probability of Xt given previous values of x. This is the entropy of the probability distribution estimated by the model.

Fixed.

– Line 667: the study by Di Liberto from which the EEG data come uses a ridge regression (ridge regularization). Is there a reason to use a non-regularized regression in this case? This should be discussed in the methods.

See our response in “Essential revisions” above.

References

Bharucha, J. J. (1987). Music cognition and perceptual facilitation: A connectionist framework. *Music Perception*, *5*, 1–30. https://doi.org/10.2307/40285384

Broderick, M. P., Anderson, A. J., Di Liberto, G. M., Crosse, M. J., and Lalor, E. C. (2018). Electrophysiological Correlates of Semantic Dissimilarity Reflect the Comprehension of Natural, Narrative Speech. *Current Biology*, *28*(5), 803-809.e3. https://doi.org/10.1016/j.cub.2018.01.080

Cohen, J. E. (1962). Information theory and music. *Behavioral Science*, *7*(2), 137–163.

https://doi.org/10.1002/bs.3830070202

Conklin, D., and Witten, I. H. (1995). Multiple viewpoint systems for music prediction. *Journal of New Music Research*, *24*(1), 51–73. https://doi.org/10.1080/09298219508570672

Heilbron, M., Armeni, K., Schoffelen, J.-M., Hagoort, P., and de Lange, F. P. (2022). A hierarchy of linguistic predictions during natural language comprehension. *Proceedings of the National Academy of Sciences*, *119*(32), e2201968119. https://doi.org/10.1073/pnas.2201968119

Huang, C.-Z. A., Vaswani, A., Uszkoreit, J., Shazeer, N., Simon, I., Hawthorne, C., Dai, A. M., Hoffman, M. D., Dinculescu, M., and Eck, D. (2018). Music Transformer. *ArXiv:1809.04281 [Cs, Eess, Stat]*.

http://arxiv.org/abs/1809.04281

Krumhansl, C. L. (2015). Statistics, Structure, and Style in Music. *Music Perception*, *33*(1), 20–31. https://doi.org/10.1525/mp.2015.33.1.20

Liberto, G. M. D., Pelofi, C., Shamma, S., and Cheveigné, A. de. (2020). Musical expertise enhances the cortical tracking of the acoustic envelope during naturalistic music listening. *Acoustical Science and Technology*, *41*(1), 361–364. https://doi.org/10.1250/ast.41.361

Meyer, L. B. (1957). *Emotion and Meaning in Music*. University of Chicago Press.

Morgan, E., Fogel, A., Nair, A., and Patel, A. D. (2019a). Statistical learning and Gestalt-like principles predict melodic expectations. *Cognition*, *189*, 23–34. https://doi.org/10.1016/j.cognition.2018.12.015

Morgan, E., Fogel, A., Nair, A., and Patel, A. D. (2019b). Statistical learning and Gestalt-like principles predict melodic expectations. *Cognition*, *189*, 23–34. https://doi.org/10.1016/j.cognition.2018.12.015 Narmour, E. (1990). *The analysis and cognition of basic melodic structures: The implication-realization model* (pp. xiv, 485). University of Chicago Press.

Narmour, E. (1992). *The Analysis and Cognition of Melodic Complexity: The Implication-Realization Model*. University of Chicago Press.

Pearce, M. T., and Wiggins, G. A. (2012). Auditory Expectation: The Information Dynamics of Music Perception and Cognition. *Topics in Cognitive Science*, *4*(4), 625–652. https://doi.org/10.1111/j.17568765.2012.01214.x

Planton, S., Kerkoerle, T. van, Abbih, L., Maheu, M., Meyniel, F., Sigman, M., Wang, L., Figueira, S., Romano, S., and Dehaene, S. (2021). A theory of memory for binary sequences: Evidence for a mental compression algorithm in humans. *PLOS Computational Biology*, *17*(1), e1008598. https://doi.org/10.1371/journal.pcbi.1008598

Schellenberg, E. G. (1997). Simplifying the Implication-Realization Model of Melodic Expectancy. *Music Perception: An Interdisciplinary Journal*, *14*(3), 295–318. JSTOR. https://doi.org/10.2307/40285723

Schmitt, L.-M., Erb, J., Tune, S., Rysop, A. U., Hartwigsen, G., and Obleser, J. (2021). Predicting speech from a cortical hierarchy of event-based time scales. *Science Advances*, *7*(49), eabi6070. https://doi.org/10.1126/sciadv.abi6070

Smith, N. J., and Kutas, M. (2015). Regression-based estimation of ERP waveforms: I. The rERP framework. *Psychophysiology*, *52*(2), 157–168. https://doi.org/10.1111/psyp.12317

Temperley, D. (2008). A Probabilistic Model of Melody Perception. *Cognitive Science*, *32*(2), 418–444. https://doi.org/10.1080/03640210701864089

Temperley, D. (2014). Probabilistic Models of Melodic Interval. *Music Perception*, *32*(1), 85–99. https://doi.org/10.1525/mp.2014.32.1.85

Wilcox, E. G., Gauthier, J., Hu, J., Qian, P., and Levy, R. (2020). *On the Predictive Power of Neural Language Models for Human Real-Time Comprehension Behavior* (arXiv:2006.01912). arXiv. https://doi.org/10.48550/arXiv.2006.01912